



**Revisiting the trend in the occurrences of the "warm Arctic-cold Eurasian continent"**
**temperature pattern**
Lejiang Yu[1,2]*, Shiyuan Zhong[3], Cuijuan Sui[4] , and Bo Sun[1]
1MNR Key Laboratory for Polar Science, Polar Research Institute of China, Shanghai, China
2 Southern Marine Science and Engineering Guangdong Laboratory (Zhuhai), Zhuhai, Guangdong,
China
3Department of Geography, Environment and Spatial Sciences, Michigan State University, East
Lansing, MI, USA
4 National Marine Environmental Forecasting Center, Beijing, China
*Corresponding Author's address
Dr. Lejiang Yu
MNR Key Laboratory for Polar Science, Polar Research Institute of China
451 Jinqiao Rd. Shanghai, 200136
Phone: 86-21-58712034,
Email: yulejiang@sina.com.cn





**Abstract.** The recent increasing trend of "warm Arctic, cold continents" has attracted much attention,
but it remains debatable as to what forces are behind this phenomenon. Here, we revisited
surface-temperature variability over the Arctic and Eurasian continent by applying the
Self-Organizing-Map (SOM) technique to gridded daily surface temperature data.   Nearly 40% of the
surface temperature trends are explained by the nine SOM patterns that depict the switch to the current
warm Arctic-cold Eurasia pattern at the beginning of this century from the reversed pattern that
dominated the 1980s and the 90s. Further, no cause-effect relationship is found between the Arctic
sea-ice loss and the cold spells in high-mid latitude Eurasian continent suggested by earlier studies.
Instead, the increasing trend in warm Arctic-cold Eurasia pattern appears to be related to the anomalous
atmospheric circulations associated with two Rossby wavetrains triggered by rising sea surface
temperature (SST) over the central North Pacific and the North Atlantic Oceans. On interdecadal
timescale, the recent increase in the occurrences of the warm Arctic-cold Eurasia pattern is a fragment
of the interdecadal variability of SST over the Atlantic Ocean as represented by the Atlantic
Multidecadal Oscillations (AMO), and over the central Pacific Ocean.
**Key words:** Warm Arctic-cold Eurasian continent, Arctic Sea ice, the Kara-Barents Sea, the
Self-Organizing-Map (SOM), the Pacific Decadal Oscillation (PDO), the Atlantic Multidecadal
Oscillation (AMO)



## 1 Introduction

In recent decades, winter season temperature in the Arctic has been rising at a rate faster than the
warming experienced in any other region of the world (Stroeve et al., 2007; Screen and Simmonds,
2010; Stroeve, 2012). In contrasts, there has been an increasing trend in colder than normal winters
over the northern mid-latitude continents (Mori et al., 2014). This pattern of opposite winter
temperature trend between the Arctic and high-mid latitude continents, referred to as the warm
Arctic-cold continents pattern (Overland et al., 2011; Cohen et al., 2014; Walsh, 2014), has also been
observed on the interannual timescale (Mori et al., 2014; Kug et al., 2015). The question as to what
processes are responsible for the opposite change of winter air temperature between the Arctic and
mid-latitudes remain open (Vihma, 2014; Barnes and Screen, 2015).
A number of studies have attributed the recent warm Arctic-cold continents pattern to the Arctic sea
ice loss (Inoue et al., 2012; Tang et al., 2013; Mori et al., 2014; Kug et al., 2015; Cohen et al., 2018;
Mori et al., 2019). Sea ice variability in different parts of the Arctic Ocean has been linked to climate
variability in different parts of the world. Specifically, sea ice loss in the Barents and Kara Seas has
been linked to cold winters over East Asia, while a similar connection has been found between cold
winters in North America and sea ice retreat in the East Siberian and Chukchi Seas (Kug et al., 2015).
A most recent study (Matsumura and Kosaka, 2019) attributed the warm Arctic-cold continents pattern
to the combined effect of Arctic sea ice loss and the atmospheric teleconnection induced by tropical
Atlantic sea-surface temperature (SST) anomalies. Some recent studies have suggested that the
mid-latitude atmospheric circulation anomalies play a role in the formation of the warm Arctic-cold
continents pattern (Luo et al., 2016; Peings et al., 2019).
Other studies, however, found no cause-and-effect relationship between Arctic sea ice loss and



mid-latitude climate anomalies (Blackport et al., 2019; Fyfe, 2019). Numerical modeling studies using
coupled ocean and atmospheric models simulated no cold mid-latitude winters when the models were
forced with reduced Arctic sea ice cover (McCusker et al., 2016; Sun et al., 2016; Koenigk et al., 2019;
Blackport et al., 2019; Fyfe, 2019). The results from these studies pointed to internal atmospheric
variability as the likely cause for cold winters in mid-latitudes. Some studies have also suggested that
on the interannual timescale mid-latitude atmospheric circulation anomalies triggered by the Pacific
and Atlantic SST oscillations may explain both the Arctic sea ice loss and the cooling of the high-mid
latitudes (Lee et al., 2011; Matsumura and Kosaka, 2019; Clark and Lee, 2019). The Gulf Stream has
also been linked to the Barents Sea ice loss and Eurasian cooling (Sato et al., 2014).
Despite the recent attention given to the warm Arctic-cold continents pattern, it remains debatable as
to what processes may be responsible for this phenomenon. In this study, we revisit surface temperature
variability over the Arctic and Eurasia continent (40-90 ˚N, 20-130 ˚E), where the warm Arctic-cold
continents pattern is a prominent feature (Cohen et al., 2014; Mori et al., 2014), by applying the
Self-Organizing-Map (SOM) technique to daily surface temperature over the recent four decades. We
will show that while the warm Arctic-cold Eurasian continent pattern has dominated the recent two
decades, its opposite pattern, cold Arctic-warm Eurasia continent, appeared frequently in the 1980s and
the 90s. Using century-long data, we will further show that the warm Arctic-cold Eurasian continent
pattern is an intrinsic climate mode and the recent increasing trend in its occurrence is a reflection of an
interdecadal variability of the pattern. Using regression method, we explain the reason for the recent
increasing occurrences of the warm Arctic-cold continents pattern. We also assess the role of the SST
anomalies over the North Pacific and Atlantic Oceans in the variability of the warm Arctic-cold Eurasia
pattern on the interdecadal time scale.





**2   Datasets and methods**
From the perspective of nonlinear dynamic, a region's climate has its intrinsic modes of variability, but
the frequency of occurrence of these internal modes can be modulated by remote forces external to the
region (Palmer, 1999l; Hoskins and Woollings, 2015; Shepherd, 2016). In this study we will first obtain
the main modes of variability of wintertime surface temperature in a region (40-90 °N, 20-130 °E) by
applying the SOM method (Kohonen, 2001) to daily surface temperature data for the 40 winters in the
1979-2019 period. The use of daily data over four decades allows for capturing the variability across
two time scales (synoptic and decadal). We will then determine, through regression and composite
analyses, the relationships of these modes of climate variability of surface air temperature to known
climate variability modes at corresponding time scales.
2.1  Datasets
Daily surface air temperature and other climate variables used in the current analyses, including 500
hPa geopotential height, 800-hPa wind and mean sea level pressure, all come from the European Centre
for Medium-Range Weather Forecasts   Re-Analysis (ERA), the interim version (ERA-Interim; Dee et
al., 2011). Compared to the earlier versions of ERA (e.g., ERA-40, Uppala et al., 2005) and other
global re-analysis products (e.g. the NCEP reanalysis, Kalnay et al., 1996), ERA-Interim has been
found to be more accurate in portraying the Arctic warming trend (Dee et al., 2011; Screen and
Simmonds, 2011) despite its known warm and moist bias in the surface layer (Jakobson et al., 2012).
Gridded monthly SST data used in the current analysis are obtained from the US National Oceanic and
Atmospheric          Administration          (NOAA)          data          archives
(ftp://ftp.cdc.noaa.gov/Datasets/noaa.oisst.v2.highres/) (Reynolds et al. 2007).
The results obtained from the data within the recent four decades are put into the context of the





variability over longer time scales using data from the Twentieth Century Reanalysis project, version
2c (20CR) that spans more than a century from 1851 through 2015 (Compo et al., 2011). The 20CR
reanalysis data has a horizontal resolution of 2 ° latitude by 2 ° longitude and temporal resolution of 6
hours. Through the assimilation of surface observational pressure data, the 20CR reanalysis was
produced by the model whose lower boundary condition is derived from monthly SST and sea ice
conditions. Various indices used to describe known modes of climate variability are obtained from
NOAA's Climate prediction Center (CPC) (https://www.esrl.noaa.gov/psd/data/climateindices/list/),
which include Arctic oscillation (AO), Northern Atlantic Oscillation (NAO), Atlantic Multidecadal
Oscillation (AMO) (Enfield et al., 2001) and PDO (Mantua et al., 1997) indices.
2.2 Methods
The 40-year, daily surface temperature over the study region (40-90 °N, 20-130 °E) is decomposed using
the SOM method. SOM is a clustering method based on neural network that can transform
multi-dimensional data into a two-dimensional array without supervised learning. The array includes a
series of nodes arranged by a Sammon map (Sammon, 1969). Each node in the array has a vector that
can represent a spatial pattern of the input data. The distance of any two nodes in the Sammon map
represents the level of similarity between the spatial patterns of the two nodes. Because SOM has fewer
limitations than most other commonly used clustering methods, (e.g., orthorgonality required by the
empirical orthogonal function or EOF method ), the SOM method can describe better the main
variability patterns of the input data (Reusch et al., 2005).
SOM method has been used in atmospheric research at mid and high latitudes of the northern
hemisphere (Skific et al., 2009; Johnson and Feldstein, 2010; Horton et al., 2015; Loikith and Broccoli,
2015; Vihma et al., 2019). For example, Johnson and Feldstein (2010) identified the spatial patterns of



the daily wintertime North Pacific sea level pressure and related the variability of the occurrences of
those patterns to some large-scale circulation indices. Loikith and Broccoli (2015) compared observed
and model-simulated circulation patterns across the North American domain. SOM method was used to
detect circulation pattern trends in a subset of North America during two periods (Horton et al., 2015).
In this study, the SOM method is applied to wintertime daily temperature anomalies obtained by
subtracting 40-year averaged daily temperature from the original daily temperature at each grid point.
Prior to SOM analysis, it is necessary to determine how many SOM nodes are needed to best capture
the variability in the data. According to previous studies (Lee and Feldstein, 2013; Gibson et al., 2017;
Schudeboom et al., 2018), the rule for determining the number of SOM nodes is that the number should
be sufficiently large to capture the variability of the data analyzed, but not too large to introduce
unimportant details. Table 1 shows the averaged spatial correlation between all daily surface air
temperature and their matching nodes. There is an increase in correlation coefficients from 0.26 for a
$3 \times 1$ grid to 0.51 for a $4 \times 4$ grid, but the gain from a $3 \times 3$ grid to a $4 \times 4$ grid is relatively small. Hence, a
$3 \times 3$ grid seems to meet the above-mentioned rule and will be utilized in this study.
The contribution of each SOM node to the trend in wintertime surface temperature is calculated by
the product of each node pattern and its frequency trend normalized by the total number of wintertime
days (90, Lee and Feldstein, 2013). The sum of the contributions from all nodes denotes the
SOM-explained trends. Residual trends are equal to the subtraction of SOM-explained trends from the
total trends. The statistical significance in this study is tested by using the Student's t test.
**3    Results**
3.1 Surface temperature variability
The majority of the 9 SOM nodes depict a dipole pattern characterized by opposite changes in surface



temperature between the Arctic Ocean and the Eurasian continent, although the sign switch does not
always occur at the continent-ocean boundary (Figure 1). The position of the boundary between the
warm and cold anomalies reflect the transition between the cold Arctic-warm Eurasia pattern (denoted,
in descent order of the occurrence frequency, by nodes 3, 9, 6), to the warm Arctic-cold Eurasia pattern
(depicted, in descent order of the occurrence frequency, by nodes 1, 7, 4). The spatial patterns
represented by the first group of nodes (3, 9, 6) are almost mirror images of the patterns denoted by the
corresponding nodes in the second group (1, 7, 4). For example, the first node in group 1 (node 9,
15.4%) and in group 2 (node 1, 17.1%) show a mirror image pattern with cold (warm) anomalies in the
Arctic Ocean extending into northern Eurasia and warm (cold) anomalies in the rest of the Eurasia
continent in the study domain. In both cases, the region of maximum anomalies is centered near
Svalbard, Norway. The second most frequent pattern, denoted by node 3 (17.2%) and 7 (13.7%) in the
two groups, respectively, has the boundary of separation moved northward from northern Eurasia
continent toward the shore of the Arctic Ocean. While the maximum anomaly in the Arctic Ocean
remains close to Svalbard, maximum values over the continent are found in central Russia. Nodes 4-6
display a noticeable transition from node 1 to node 7 and from node 3 to node 9, respectively. Although
nodes 2 and 8 show an approximate monopole spatial pattern, they also represent a transition between
nodes 1 and 3, and between nodes 7 and 9, respectively. Above SOM analysis cannot consider the trend
in surface air temperature. The result is similar while removing the trend (Not shown).

The temporal variability on this time scale is typically related to synoptic processes and hence the

questions are what synoptic patterns are responsible for the occurrence of the spatial patterns depicted
by each of the 9 SOM nodes and how these patterns are related to those of the Arctic sea ice anomalies?
These questions can be answered by using the composite method. Specifically, for each node,



composite maps are made respectively for the anomalous 500-hPa geopotential height, mean sea level
pressure, 850-hPa wind, downward longwave radiation, surface turbulent heat flux, and sea ice
concentration over all the days when the spatial variability of the surface temperature anomalies is best
matched by the spatial pattern of that node.
3.2 Large-scale circulation patterns
For all nodes, the spatial pattern of the composited 500 hPa-geopotential height anomalies (Figure 2) is
similar to that of mean sea level pressure anomalies (Not shown), indicating an approximately
barotropic structure. For nodes 1, 4 and 7, 500-hPa height anomalies show a dipole structure of positive
values over Siberia and negative values to its south. Anomalous southwesterly winds on the western
side of the anticyclone over Siberia transport warm and moist air from northern Europe and the North
Atlantic Ocean into the Atlantic sector of the Arctic Ocean (Figure 3), providing a plausible
explanation of the warm surface temperature anomalies in the region (Figure 1). On the eastern side of
the anticyclone, anomalous northwesterly winds bring cold and dry air from the Arctic Ocean into
Eurasia continent, which is consistent with the negative surface temperature anomalies there. The
opposite occurs for nodes 3, 6 and 9. A similar explanation involving anomalous pressure and wind
fields can be applied to other nodes. The dipole structure that dominates the anomalous 500-hPa height
fields over the North Atlantic Ocean for most nodes resembles the spatial pattern of the NAO. In
addition, the patterns for a few nodes, such as nodes 4 and 7, have some resemblance to the spatial
pattern of the AO over larger geographical region. The possible connection to NAO and AO is further
investigated by averaging the daily index values of NAO or AO over all occurrence days for each node.
The results (Table 2) show that nodes 1, 2, 3 (5, 8, 9) correspond to a significant positive (negative)
phase of the NAO index characterized by negative (positive) height anomalies over Iceland and





positive (negative) values over the central North Atlantic Ocean. Association is also found between
nodes 1, 2, 3, and 6 (5, 7, 8, and 9) and the positive (negative) phases of the AO index.
3.3 Downward radiative fluxes
Besides the anomalous circulation patterns, anomalous surface radiative fluxes may also play a role in
shaping the spatial pattern of surface temperature variability. In fact, the spatial pattern of the mean
anomalous daily downward longwave radiation for an individual node (Figure 4) is in good agreement
with the spatial pattern of the surface temperature anomalies of that node. In other words, increased
downward longwave radiation is associated with positive surface temperature anomalies, and vice
versa. As expected from previous studies (e.g., Sedlar et al. 2011), there is a significant positive
correlation between downward longwave radiative fluxes and the anomalous total column water vapor
and mid-level cloud cover (not shown). The correlation to low- and high-level cloud cover is, however,
not significant (Not shown). Most of the water vapor in both the Arctic and Eurasia is derived from the
North Atlantic Ocean, but the water vapor is transported into the Arctic by southwesterly flows and into
Eurasia by northwesterly winds. The anomalous shortwave radiation corresponding to each node (not
shown) is an order of magnitude smaller that of the longwave radiation anomalies and has a spatial
pattern opposite to that of the mid-level cloud cover and the longwave radiation anomalies.
3.4 Sea ice
The analyses presented above attempt to explain the spatial pattern of surface temperature variability
for each node from the perspective of anomalous heat advection and surface radiative fluxes. As
mentioned earlier, there has been a debate in the literature about the role played by the sea ice
anomalies in the Barents and Kara Seas in the development of the warm Arctic-cold Eurasia pattern.
Here, we examine the anomalous turbulent heat flux (Figure 5) and sea ice concentration (Figure 6) for





each node. Turbulent heat flux is considered positive when it is directed from the atmosphere
downward to the ocean or land surfaces. Thus, a positive anomaly indicates either an increase in the
atmosphere-to-surface heat transfer or a decrease in the heat transfer from the surface to the atmosphere.
The magnitude of anomalous turbulent heat flux is found to be comparable to that of anomalous
downward longwave radiation (Figure 4). For all nodes, the heat flux anomalies are larger over ocean
than over land. For node 1, positive turbulent heat flux anomalies occur mainly over the Barents Sea,
the western and central North Atlantic Ocean and the eastern North Pacific Ocean, indicating an
increase in heat transport from the air to the ocean due possibly to an increase in vertical temperature
gradient caused by warm air advection associated with anomalous circulation. The downward heat
transfer results in sea ice melt in the Greenland Sea and the Barents Sea (Figure 6). For node 4, the
anomalous southerly winds over the Nordic Sea produce larger positive turbulent heat flux anomalies.
For node 7, the anticyclone is located more northwards, which generates opposite anomalous winds
between the Nordic and northern Barents Seas and the southern Barents Sea and thus opposite turbulent
heat flux anomalies that are consistent with the opposite sea ice concentration anomalies in the two
regions. For nodes 3, 6, and 9, the anomalous cold air from the central Arctic Ocean flows into warm
water in the Nordic and Barents Seas, producing negative turbulent heat flux anomalies and positive
sea ice concentration anomalies. Sorokina et al. (2016) noted that turbulent heat flux usually peaks 2
days before changes in surface temperature pattern occur. The pattern of the composted anomalous
turbulent heat flux 2 days prior to the day when the nodes occur (not shown) is similar to the
current-day pattern in Figure 6. Our results support the conclusion of Sorokina et al. (2016) and
Blackport et al. (2019) that the anomalous atmospheric circulations lead to the anomalous sea ice
concentration in the Barents Sea.



3.5 Contributions of SOM nodes to the trends in wintertime surface temperature
The results above suggest that both the surface temperature anomaly patterns over the Arctic Ocean and
Eurasian continent and the sea ice concentration anomalies in the Nordic and Barents Seas can be
explained largely by changes in atmospheric circulations and the associated vertical and horizontal heat
and moisture transfer by mean and turbulent flows. Next, we assess the contributions of these nodes to
the trend in wintertime surface temperature.

We first examine the time series of the accumulated number of days for each node in each winter for

the 1979-2019 period (Figure 7). The time series for nodes 1, 4, 6, and 9 exhibit variability on
interannual as well as decadal time scales. The occurrence frequency is noticeably larger after 2003
than prior to 2003 for nodes 1 and 4, and vice versa for nodes 6 and 9, and the difference between the
two periods is significant at 95% confidence level. Given the spatial patterns of these four nodes
(Figure 1), this indicates that the warm Arctic-cold Eurasia pattern occurred more frequently after 2003.
A linear trend analysis of the time series for each node (Table 2) reveals significant positive trends in
occurrence frequency for nodes 1 and 4 and significant negative trends for nodes 6 and 9, which agree
with the result from a previous study (Clark and Lee, 2019) that suggested an increasing trend of the
warm Arctic and cold Eurasia pattern.

These trends in the occurrence frequency of the SOM nodes contribute to the trends in the total

wintertime (DJF) surface temperature anomalies (Figure 8, top panel) that have significant positive
trends over the Arctic Ocean and in regions of Northern and Southern Europe and negative trends in
Central Siberia. The contribution, however, varies from node to node (Figure 9). Node 1 has the largest
domain-averaged contribution of 18.7%, followed by its mirror node (node 9) at 10.1%. Nodes 4 and 6
account for 2.8% and 4.3% of the total trend, respectively. None of the remaining nodes explain more



than 2%. All nodes together explain 39.5% of the total trend in wintertime surface air temperature. The
spatial pattern of the SOM-explained trends (Figure 8, middle panel) is similar to the warm
Arctic--cold continent pattern, whereas the residual trend resembles more the total trend (Figure 8
bottom panel).
3.6 Mechanisms
The results presented above indicate that the SOM patterns explain nearly 40% of the trend in
wintertime surface air temperature anomalies and majority of the contributions (35 out of 40%) come
from the two pairs of the nodes (nodes 1, 9, and 4, 6).   The analyses hereafter will focus on these four
nodes. Below we assess the atmospheric and oceanic conditions associated with the occurrences of the
four nodes via regression analysis. Specifically, the anomalous seasonal SST and atmospheric
circulation variables are regressed onto the normalized time series of the number of days when each of
the four nodes occurs (Figures 10, 11, and 12).
For node 1, the SST regression pattern in the Pacific Ocean shows significant positive anomalies
over the tropical western Pacific Ocean and central North Pacific Ocean. The positive SST anomalies
also occur over most of the North Atlantic. Negative SST anomalies occur over the central tropical
Pacific Ocean, though they are not significant at 95% confidence level. The SST regression pattern is
reversed for node 9. The corresponding anomalous 500-hPa height regression shows two Rossby
wavetrains: one is excited over the central Pacific Ocean and propagates northeastwards into North
America and North Atlantic Ocean, and the other, which displays the stronger signal, originates from
central North Atlantic and propagates northeastwards to the Arctic Ocean and southeastwards to the
Eurasian continent and the western Pacific Ocean. The large SST anomalies over the Nordic Ocean
augment the wave signal through local air-sea interaction. The wave activity flux and streamfunction



exhibit well the horizontal propagating direction of the planetary wave. For node 9, the corresponding
anomalous 500-hPa height and streamfunction show an opposite pattern, but the wave activity flux is
similar to that of node 1.

For node 4, the SST anomalies over the tropical Pacific Ocean appear to be in a La Niña state, which

shows stronger negative SST anomalies over the eastern tropical Pacific Ocean than those for node 1.
The positive SST anomalies over the North Pacific shift more northwards relative to that of node 1. The
positive SST anomalies over the North Atlantic are weaker than those for node 1. The corresponding
wavetrain over the Pacific Ocean is stronger than that over the Atlantic Ocean, which can also be
observed in the pattern of wave activity and streamfunction. The corresponding pattern for node 6 is
nearly reversed, but there are some noticeable differences in the amplitude of the wavetrain and SST
anomalies. For example, the magnitude of the anomalous SST and the 500-hPa height over the central
North Pacific is larger for node 6 than that for node 4.

Besides the above-mentioned variables, similar regression analysis is also performed for the

anomalous 850-hPa wind field and anomalous downward longwave radiation (Not shown). Their
regression patterns, which are similar to those in Figures 3 and 4, explain well the decadal variability of
the number of days for nodes 1, 4, 6, and 9. Together, these results indicate that the decadal variability
of the occurrence frequency of the four nodes in recent decades is related to two wavetrains induced by
SST anomalies over the central North Pacific Ocean and the North Atlantic Ocean. The aforementioned
SST regression patterns over the Atlantic and Pacific Oceans also show features of the AMO and PDO
(Figure 10). Since both the AMO and PDO exhibited a phase change in the late 1990s (Yu et al., 2017),
the question is whether a similar change in the SOM frequency also appear in the late 1990s. A
comparison of the averaged frequency before and after 1998 shows a significant drop in frequency for





nodes 6 and 9 and an increase in frequency for node 1. This result suggests that the change in the AMO
and PDO indices may contribute to the change in the frequencies of the warm Arctic-cold Eurasia
continent pattern.
3.7 Interdecadal variability
The four-decade-long ERA-Interim reanalysis is not adequate for examining interdecadal to
multi-decadal variations represented by the PDO and AMO indices. Further analysis is performed using
the 20CR daily reanalysis data for the 1854-2014 period. Before applying the SOM technique to the
20CR data, we first remove the trend to eliminate the influence from the global warming. No low-pass
filter is applied before SOM analysis in order to test the stability of the SOM results for the different
periods. The spatial SOM patterns from the de-trended century-long 20CR data (Figure 13) are similar
to those for the 1979-2019 period (Figure 1). Nodes 1, 4, and 7 correspond to the positive phase of the
warm Arctic-cold Eurasia pattern and the negative phase can be observed in nodes 3, 6, and 9. The
magnitude is smaller compared to the recent four decades. The occurrence frequencies of all the nodes
(Figure 14) are close to those for the recent four decades. It indicates that the SOM method can obtain
stably the main modes of wintertime surface air temperature variability. For the recent four decades, the
time series of the number of days also displays a noticeable increasing (decreasing) trend for nodes 1
and 4 (6 and 9), suggesting that the trend in the recent four decades is a reflection of an interdecadal
variability of wintertime surface air temperature.

Next, we apply a 40-year low-pass filter to the time series of the occurrence frequencies for nodes 1,

4, 6 and 9 and the AMO and PDO indices and calculate correlations. There is a significant correlation
between the time series and the AMO index, with correlation coefficients of 0.36 for node 1, 0.27 for
node 4, -0.37 for node 6, and -0.20 for node 9, all of which are at the 95% confidence level. No



significant correlations, however, are found between the filtered time series and the PDO index. If we
define an SST index to represent the variability of SST anomalies over the central North Pacific Ocean
(20 °N-40 °N, 150 °E-150 °W), the 40-year low-pass filtered central North Pacific Ocean SST index is
now significantly correlated with the filtered time series of occurrence frequencies for nodes 1 and 9
(0.55 for node 1 and -0.46 for node 9). The results are consistent with the SST regression map for the
recent decades (Figure 10).
To confirm the effect of SST anomalies on the warm Arctic -cold Eurasia pattern, we also perform
EOF analysis of wintertime detrended seasonal surface air temperature anomalies for the 1854-2014
period (Figure 15). The spatial patterns of the first and second EOF modes show the negative phase of
the warm Arctic-cold Eurasia pattern and the 40-year low-pass filtered time series is inversely
correlated with the 40-year low-pass filtered wintertime AMO index (-0.46 $p<0.05$ for mode 1 and
-0.44 $p<0.05$ for mode 2). The 40-year low-pass filtered time series of the two EOF modes has a
significant negative correlation with the 40-year low-pass filtered central North Pacific Ocean SST
index, with correlation coefficients of -0.19 and -0.26 ($p<0.05$). Only PC1 has a significant correlation
with the PDO index (0.38 $p<0.05$). Thus, the increase in the occurrence of the warm Arctic-cold
Eurasia pattern in the recent decades is a part of the interdecadal variability of the pattern, which is
influenced by the AMO index and the central North Pacific SST.
**4 Conclusions and Discussions**
In this study, we examine the variability of wintertime surface air temperature in the Arctic and the
Eurasian continent (20 °E-130 °E) by applying the SOM method to daily temperature from the gridded
ERA-Interim dataset for the period 1979-2019 and from the 20CR reanalysis for the period 1854-2014
and the EOF method to seasonal temperature from the 20CR reanalysis for the period 1854-2014.



The spatial pattern in the surface temperature variations in the study region, as revealed by the nine
SOM nodes, is dominated by concurrent warming in the Arctic and cooling in Eurasia, and vice versa.
The nine SOM patterns explain nearly 40% of the trends in wintertime surface temperature and 88% of
that are accounted for by only four nodes. Two of the four nodes (nodes 1 and 4) represent the warm
Arctic-cold Eurasian pattern and the other two (nodes 6 and 9) depict the opposite cold Arctic-warm
Eurasia pattern. There is a clear shift in the frequency of the occurrence of these patterns near the
beginning of this century, with the warm Arctic – cold Eurasia pattern dominating since 2003, while the
opposite pattern prevailing from the 1980s through the 1990s. The warm Arctic-cold Eurasia pattern is
accompanied by an anomalous high pressure and anticyclonic circulation over the Eurasian continent.
The anomalous winds and the associated temperature and moisture advection interact with local
longwave radiative forcing and turbulence to produce positive (negative) temperature anomalies in the
Arctic (Eurasian continent). The circulation is reversed for the cold Arctic-warm Eurasia pattern. The
warm, moist air mass advected to the Arctic by the anomalous atmospheric circulations and the
increased downward turbulent heat flux also explain sea ice melt in the Barents and Kara Seas. In other
words, the sea ice loss in the Barents and Kara Seas and the cooling of the Eurasian continent can both
be traced to anomalous atmospheric circulations.
Increasing occurrences of the warm Arctic-cold Eurasian continent pattern appear to relate to rising
SST over the central North Pacific and North Atlantic Oceans (positive AMO phase). The SST
anomalies trigger two Rossby wavetrains spanning from the North Pacific Ocean, North America, and
the North Atlantic Ocean to the Eurasian continent. The two wavetrains are strengthened through local
sea-atmosphere-ice interactions in mid-high latitudes, which influence the change in the occurrence
frequency of the warm Arctic-cold Eurasian continent pattern. Our results agree with those of previous





studies (Lee et al., 2011; Sato et al., 2014; Clark and Lee, 2019). But previous studies only focus on the
effect of SST anomalies over either North Pacific or North Atlantic Oceans. We also note that the two
wavetrains excited by SST anomalies over different oceans differ in amplitudes, leading to somewhat
different warm Arctic-cold Eurasia patterns.
Using century-long data, we show that the warm Arctic-cold Eurasia pattern is an intrinsic climate
mode, which has been stable since 1854. The recent increasing trend in its occurrence is a reflection of
an interdecadal variability of the pattern resulting from the interdecadal variability of SST anomalies
over the central Pacific Ocean and over the Atlantic Ocean represented by the AMO index. Sung et al.
(2018) investigated interdecadal variability of the warm Arctic and cold Eurasia pattern and considered
the variability of the SST over the North Atlantic as its origin. Our results suggest that the variability of
the SST over the North Pacific also plays an important role. However, internal atmospheric variability
remains another potential source. The Rossby wavetrains also lead to deepening of a trough in East
Asia and generate an anomalous low and cold temperature in northern China, which further suggests
that the relationship between a warmer Arctic, especially warmer Barents and Kara Seas, and the
occurrence of cold spells in East Asia may not be as strong as previously thought (Kim et al., 2014;
Mori et al., 2014; Kug et al., 2015; Overland et al., 2015).
Our results help broaden the current understanding of the formation mechanisms for the warm
Arctic-cold Eurasia pattern. The SST anomalies over Northern Hemisphere oceans may offer a
potential for predicting its occurrence.
**Data Availability**
All data used in the current analyses are publicly available. The monthly sea ice concentration data are
available from the National Snow and Ice Data Center (NSIDC) (http://nsidc.org/data/NSIDC-0051), the
ERA-Interim reanalysis data are available from the European Center for Mid-Range Weather



Forecasting (https://www.ecmwf.int/en/forecasts/datasets/reanalysis-datasets/era-interim) and the sea
surface temperature data are available from the Hadley Centre for Climate Prediction and Research
(ftp://ftp.cdc.noaa.gov/Datasets/noaa.oisst.v2.highres/). The long-term SST data are derived from
from the Twentieth Century Reanalysis project, version 2c (20CR)
(https://climatedataguide.ucar.edu/climate-data/noaa-20th-century-reanalysis-version-2-and-2c).
**Competing interests**
The authors declare that they have no conflict of interest.
**Author Contributions**
L. Yu designed the study, with input from S. Zhong, and carried out the analyses. L. Yu and S. Zhong
prepared the manuscript. C. Sui plotted a part of Figures.
**Acknowledgements** We thank the European Centre for Medium-Range Weather Forecasts (ECMWF)
for the ERA-Interim data. This study is financially supported by the National Key R&D Program of
China (2019YFC1509102; 2017YFE0111700) and the National Natural Science Foundation of China

(41922044).
















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





Table 1. Spatial correlations (Corrs) between the daily winter (DJF) surface air
temperature and the corresponding SOM pattern for each day from 1979 to 2018.

|      | 3×1 | 2×2 | 3×2 | 4×2 | 3×3 | 5×2 | 4×3 | 5×3 | 4×4 |
|------|-----|-----|-----|-----|-----|-----|-----|-----|-----|
| Corr | 0.26 | 0.43 | 0.48 | 0.48 | 0.50 | 0.49 | 0.50 | 0.51 | 0.51 |












Table 2. Averaged anomalous NAO and AO indices for all occurrences of each SOM
node. Asterisks indicate the above 95% confidence level.

|  | Node1 | Node2 | Node3 | Node4 | Node5 | Node6 | Node7 | Node8 | Node9 |
|---|---|---|---|---|---|---|---|---|---|
| NAO | 0.38* | 0.22* | 0.12* | 0.05 | -0.22* | -0.02 | -0.07 | -0.31* | -0.32* |
| AO | 0.44* | 0.38* | 1.03* | -0.42 | -0.62* | 0.22* | -0.44* | -1.11* | -0.41* |






Table 3. Trends in the frequency of occurrences for each SOM node (day yr$^{-1}$).
Asterisks indicate the above 95% confidence level.

|       | Node1 | Node2 | Node3 | Node4 | Node5 | Node6 | Node7 | Node8 | Node9 |
|-------|-------|-------|-------|-------|-------|-------|-------|-------|-------|
| Trend | 0.80* | 0.10  | -0.18 | 0.22* | -0.02 | -0.39* | 0.17 | -0.17 | -0.50* |






Table 4. Frequencies of occurrence (%) of wintertime surface air temperature patterns
in Figure 1 for all winters before 1998 and after 1998 for the period 1979-2019.
Values with Asterisks are significantly different from climatology above the 95%
confidence level.

| SOM patterns | Frequencies of occurrence | | |
| --- | --- | --- | --- |
| | All winters | Winters before 1998 | Winters after 1998 |
| Node 1 | 17.1 | 7.4* | 26.8 |
| Node 2 | 4.4 | 3.3 | 5.4 |
| Node 3 | 17.2 | 18.8 | 15.6 |
| Node 4 | 8.6 | 5.4 | 11.7 |
| Node 5 | 3.4 | 3.4 | 3.5 |
| Node 6 | 10.2 | 15.2* | 2.1* |
| Node 7 | 13.7 | 10.6 | 16.8 |
| Node 8 | 10.1 | 12.1 | 8.0 |
| Node 9 | 15.4 | 23.7* | 7.1* |






## Figure Captions


Figure 1. Spatial patterns of SOM nodes for daily wintertime (December, January, and
February) surface air temperature anomalies (°C). The number in brackets denotes the
frequency of the occurrence for each node.
Figure 2. Corresponding 500-hPa geopotential height anomalies (gpm) for each SOM
node. Dotted regions indicate the above 95% confidence level.
Figure 3. The same as Figure 2, but for anomalous 850-hPa wind field ($ms^{-1}$).
Figure 4. The same as Figure 2, but for anomalous downward longwave radiation ($10^5$
$W\,m^{-2}$).
Figure 5. The same as Figure 2, but for anomalous turbulent heat flux (sensible and
latent heat) ($10^5\,W\,m^{-2}$). Positive values denote heat flux from atmosphere to ocean
and land and vice versa.
Figure 6. The same as Figure 2, but for anomalous sea ice concentration.
Figure 7. Time series of the number of days for occurrence of each SOM node in
Figure 1.
Figure 8. Total (top), SOM-explained (middle), and residual (bottom) trends in
wintertime surface air temperature ($^{o}C\,yr^{-1}$). Dots in the top panel indicate above 95%
confidence level.
Figure 9. Trends in surface air temperature explained by each SOM node ($°C\,yr^{-1}$).
The percentage in the upper of each panel indicates the fraction of the total trends
represented by each node.
Figure 10. Anomalous SST (°C) regressed into the normalized time series of





occurrence number for nodes 1, 4, 6, and 9.
Figure 11. As in Figure 10, but for the anomalous 500-hPa geopotential height (gpm).
Figure 12. The anomalous wave activity flux (vectors) and stream function (colors,
$10^7$ m$^2$/s) regressed onto the normalized time series of occurrence number for nodes 1,
4, 6, and 9.
Figure 13. Spatial patterns of the SOM nodes for daily wintertime (December, January,
and February) surface air temperature anomalies (℃) for the 1851-2014 period. The
number in brackets denotes the frequency of the occurrence for each node.
Figure 14. Time series of the number of days for occurrence of each SOM node in
Figure 1.
Figure 15. The (a) leading pattern and (b) its time series (PC1 and PC2) of EOF
analysis of wintertime surface air temperature anomalies. Prior to EOF analysis,
surface air temperature data are detrended. A 40-yr low-pass filtered is applied to the
time series of PC1, PC2, AMO and PDO indices.











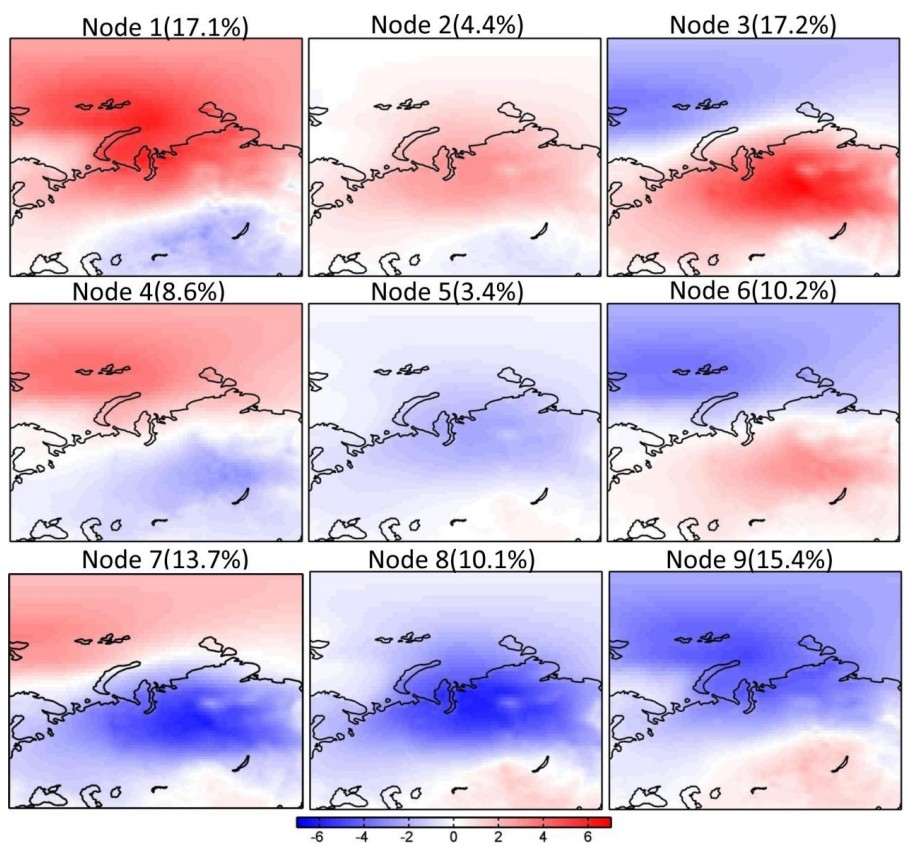


Figure 1. Spatial patterns of SOM nodes for daily wintertime (December, January, and February) surface air temperature anomalies (℃). The number in brackets denotes the frequency of the occurrence for each node.

















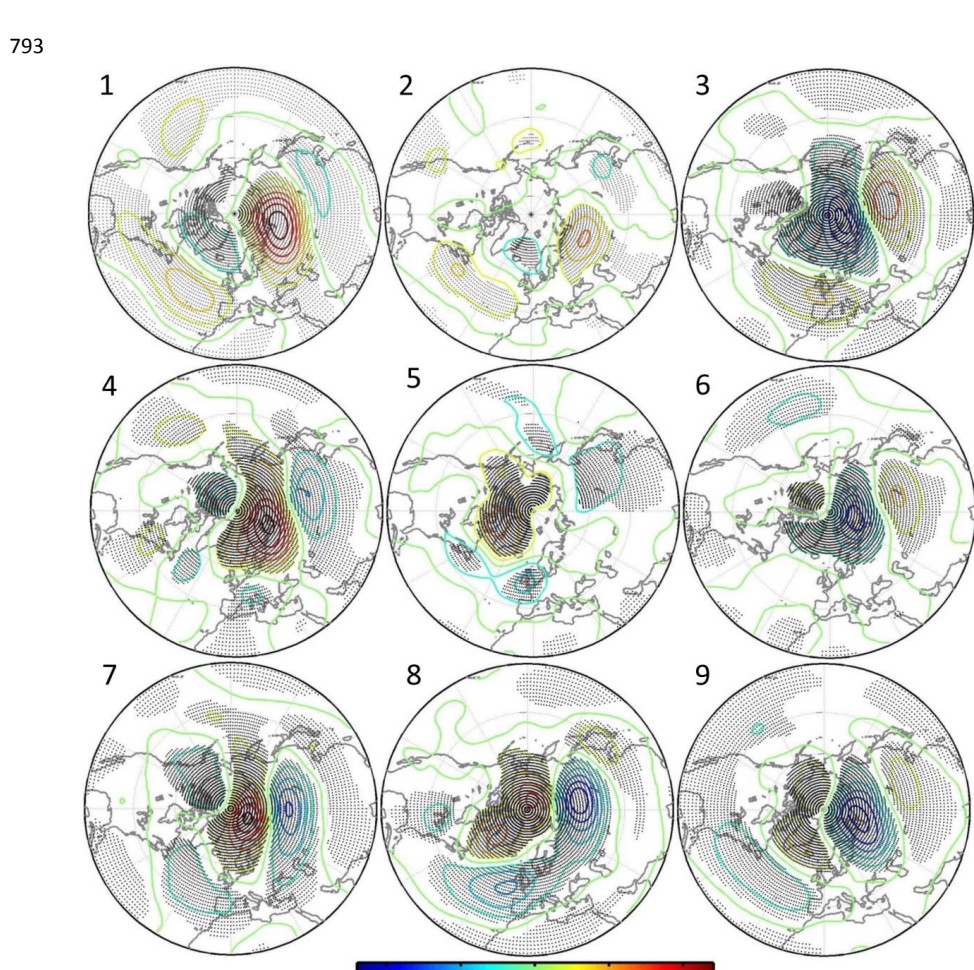


Figure 2. Corresponding 500-hPa geopotential height anomalies (gpm) for each node. Dotted
regions indicate the above 95% confidence level.





















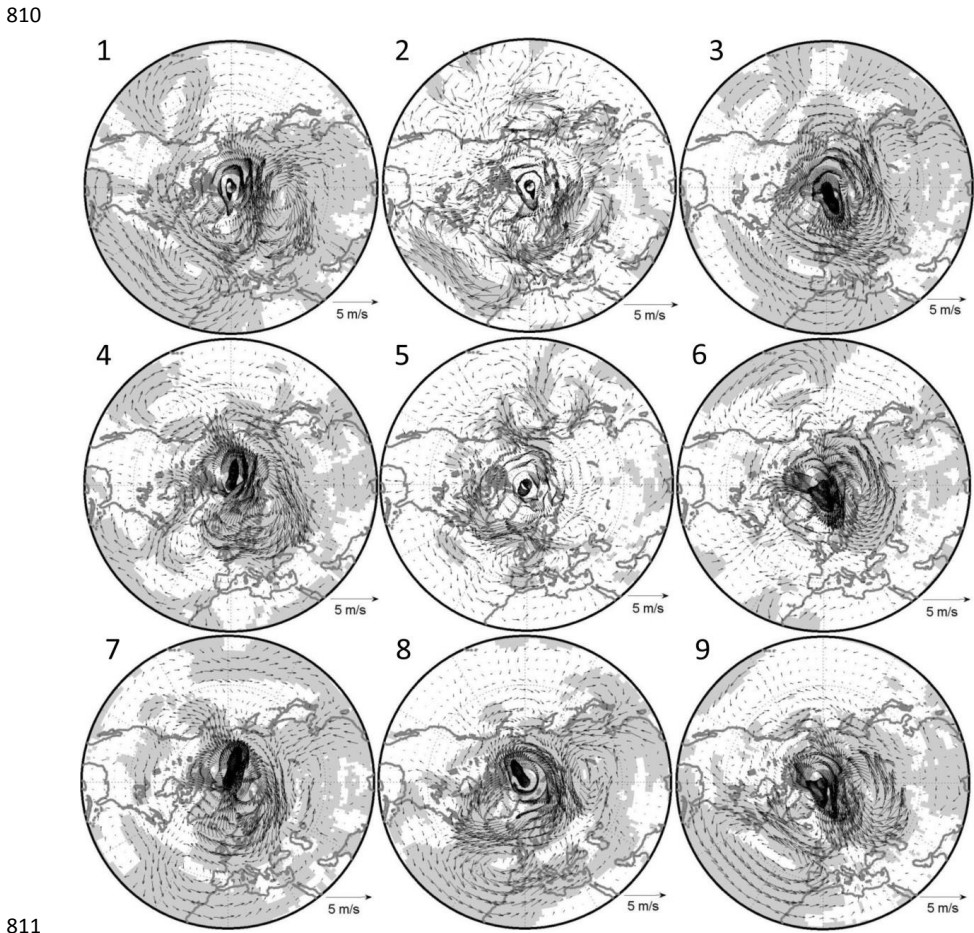

Figure 3. The same as Figure 2, but for anomalous 850-hPa wind field (ms[-1]).


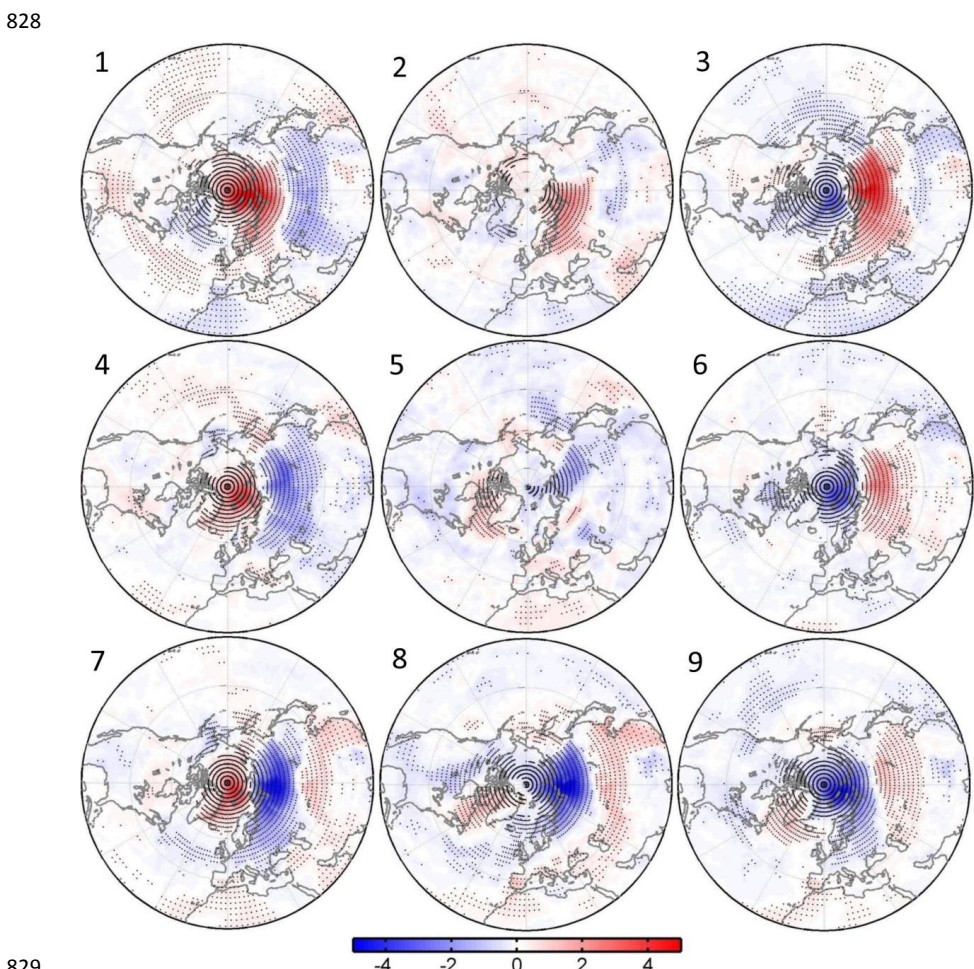


Figure 4. The same as Figure 2, but for anomalous downward longwave radiation ($10^5$ W m$^{-2}$).















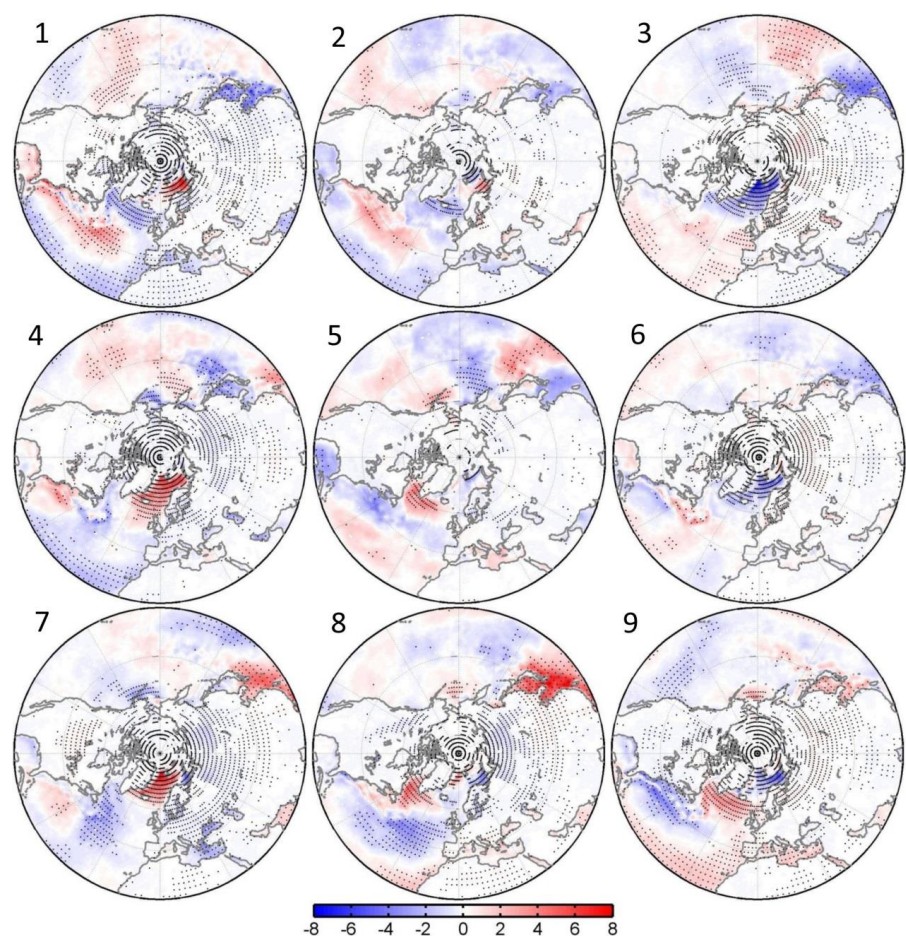


Figure 5. The same as Figure 2, but for anomalous turbulent heat flux (sensible and latent heat)
($10^5$ W m$^{-2}$). Positive values denote heat flux from atmosphere to ocean and vice versa.
















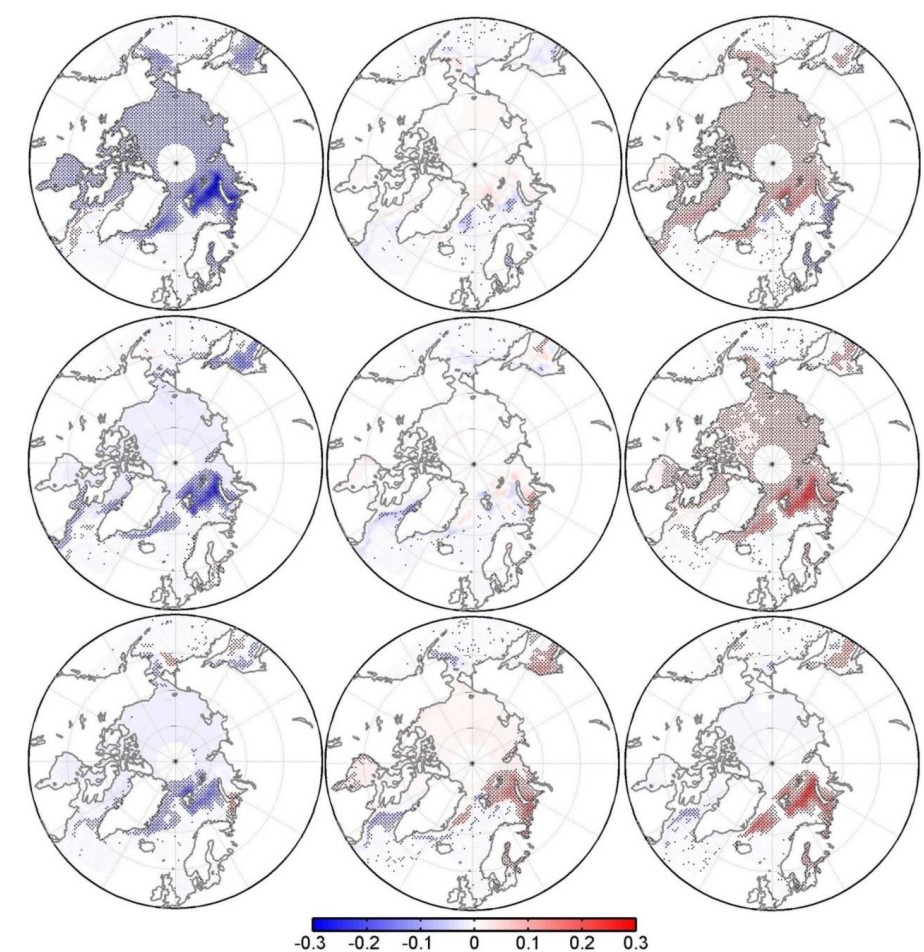

Figure 6. The same as Figure2, but for anomalous sea ice concentration.

















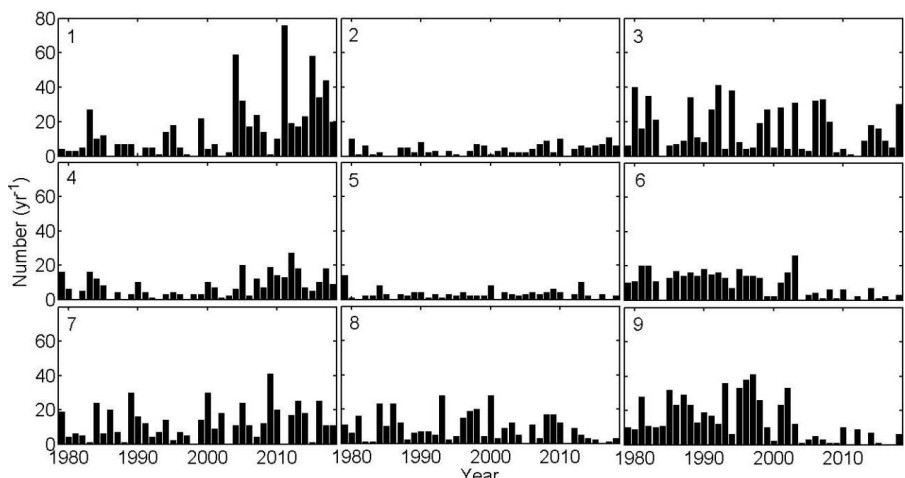


Figure 7. Time series of the number of days for occurrence of each SOM node in Figure 1.






























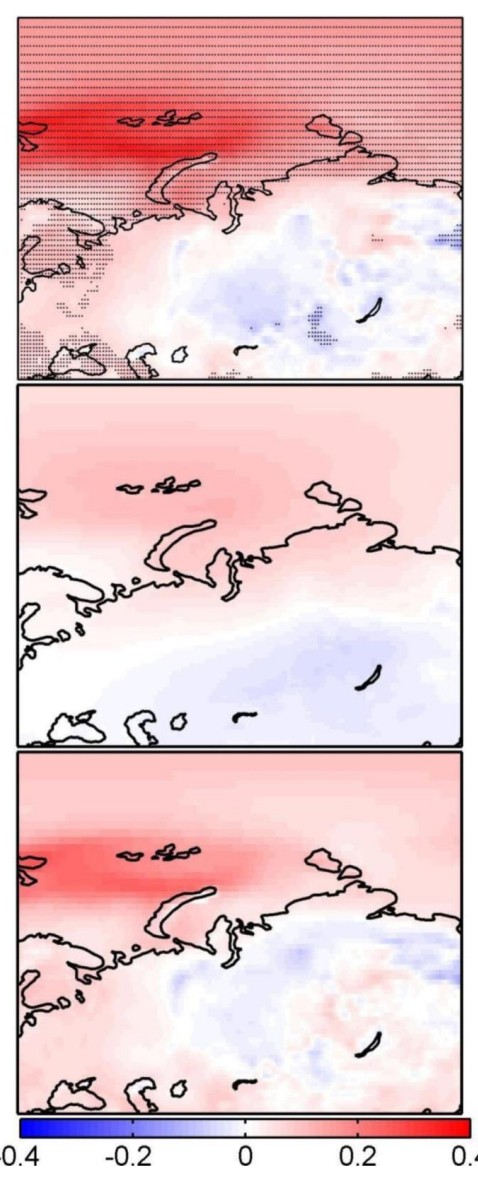


Figure 8. Total (top), SOM-explained (middle), and residual (bottom) trend in wintertime (DJF)
surface air temperature ($^{o}$ C yr$^{-1}$). Dots in the top panel indicate above 95% confidence level.



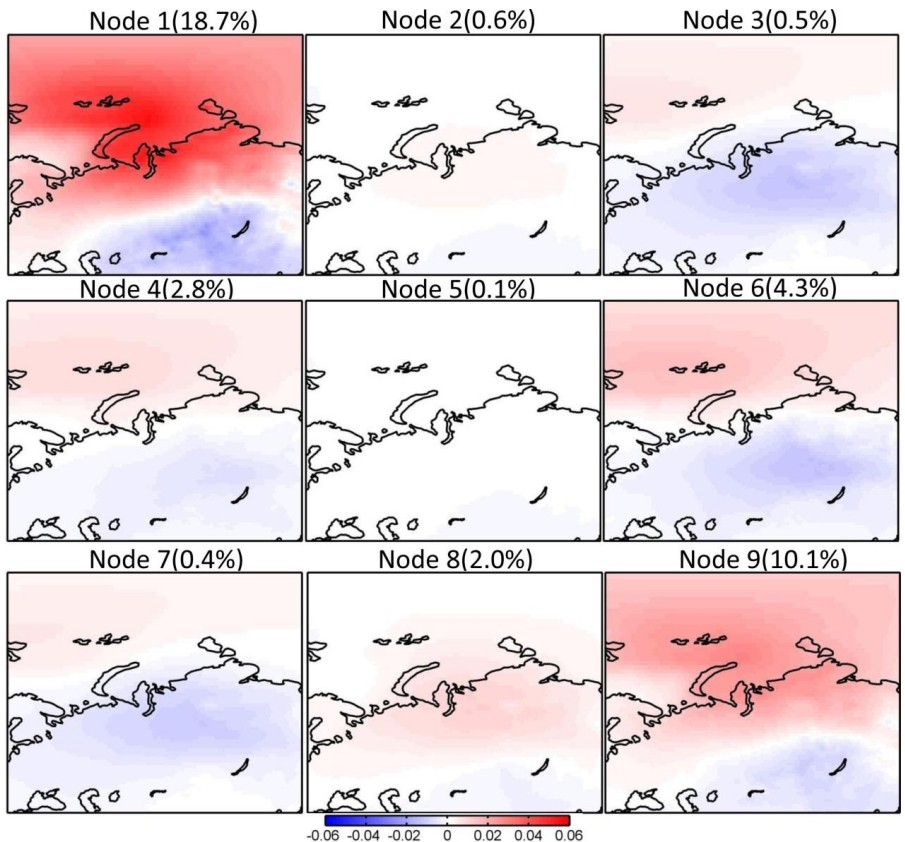


Figure 9. Trends in surface air temperature explained by each SOM node ( ℃ yr$^{-1}$). The percentage
in the upper of each panel indicates the fraction of the total trend represented by each node.

















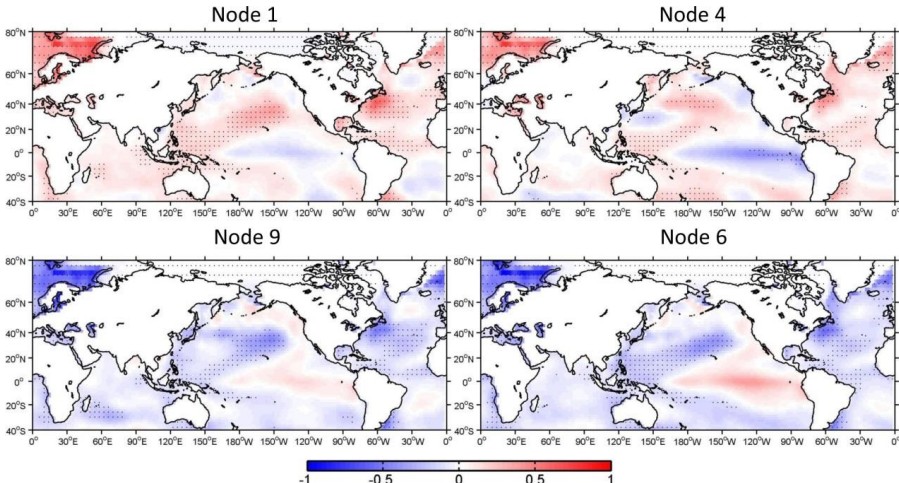


Figure 10. Anomalous SST (℃) regressed into the normalized time series of occurrence number
for nodes 1, 4, 6, and 9.




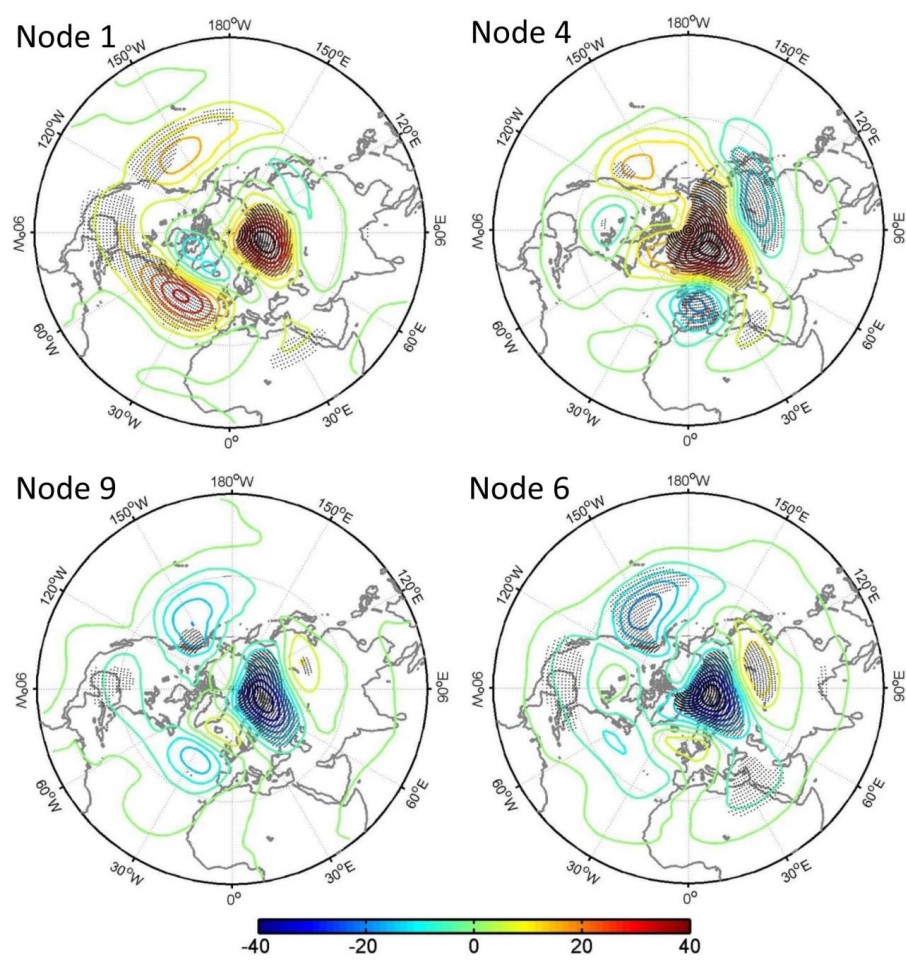


Figure 11. As in Fig. 10, but for the anomalous 500-hPa geopotential height (gpm).

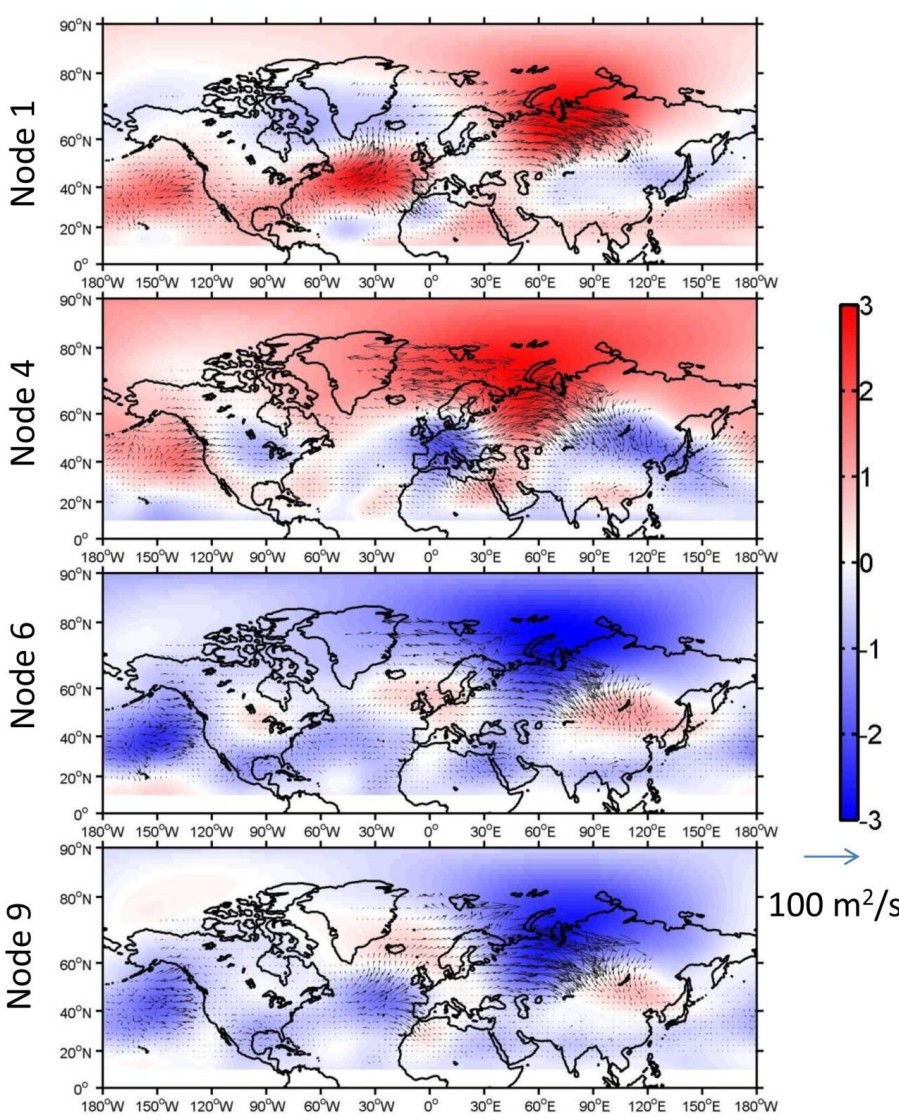


Figure 12. The anomalous wave activity flux (vectors) and stream function (colors, units: $10^7$ $m^2$/s)


regressed onto the normalized time series of occurrence number for nodes 1, 4, 6, and 9.









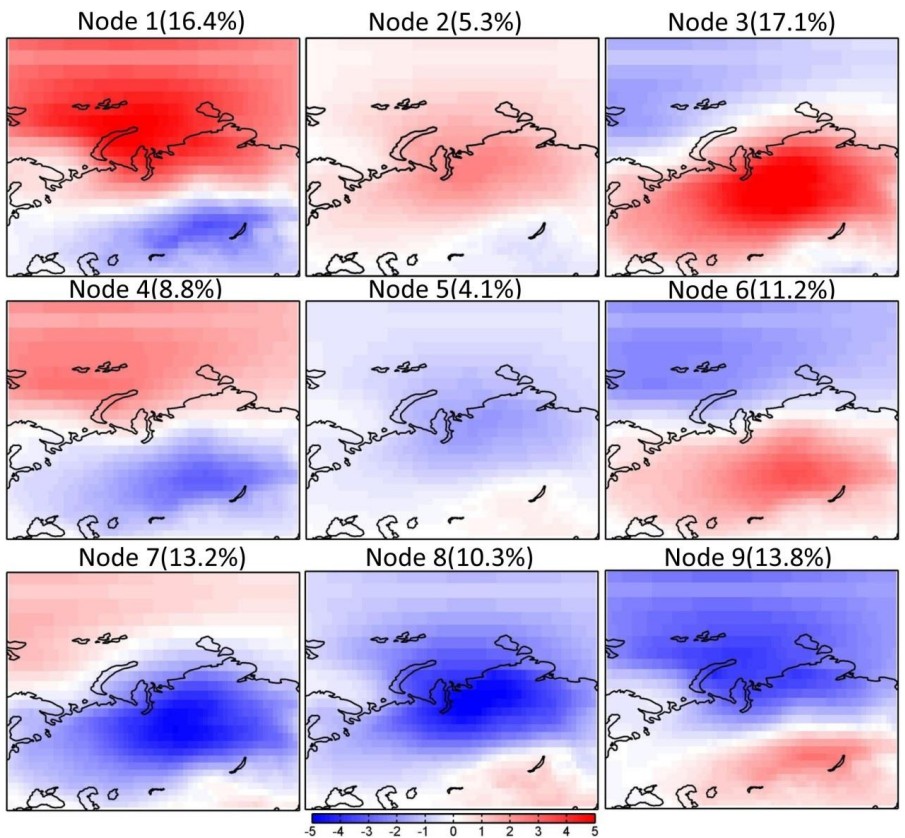


Figure 13. Spatial patterns of SOM nodes for daily wintertime (December, January, and February)
surface air temperature anomalies (℃) for the 1851-2014. The number in brackets denotes the
frequency of the occurrence for each node.



















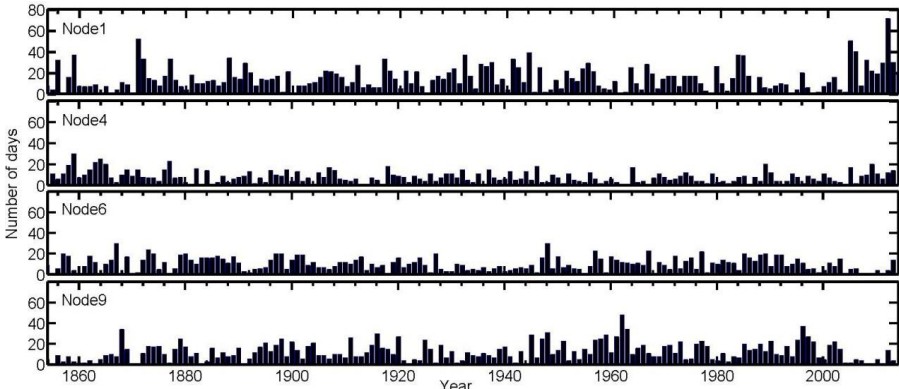


Figure 14. Time series of the number of days for occurrence of each SOM node in Figure 13.































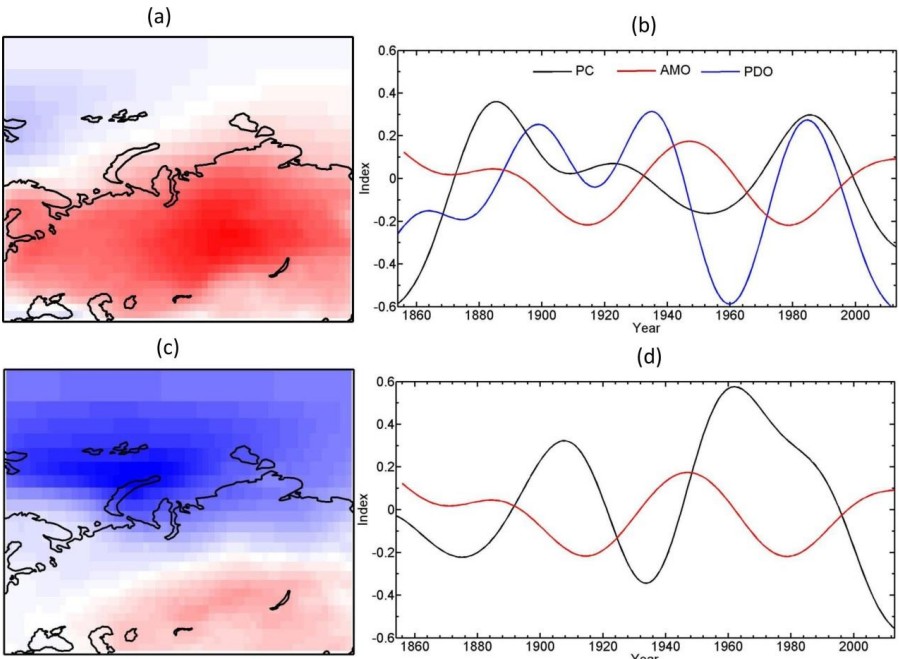

Figure 15. The (a) leading pattern and (b) its time series (PC1 and PC2) of EOF analysis of
wintertime surface air temperature anomalies. Prior to EOF analysis, surface sir temperature data
are detrended. A 40-yr low-pass filtered is applied to the time series of PC1, PC2, AMO and PDO
indices.