# Peer review of "Revisiting the trend in the occurrences of the “warm Arctic-cold Eurasian continent”"

_Atmospheric Chemistry and Physics, 2020_

## Referee Comment (RC1) · Anonymous Referee #1 · 28 Aug 2020

The linkage between the warn Arctic and mid-latitude weather and climate is a hot topic for cryosphere research community and for this reason, I see this study is interesting and worth to be noticed as a scientific publication. The manuscript is well structured, and the objectives of this study are clear. The content fits well the scope of ACP. I recommend this manuscript to be published in ACP. However, I see there are some aspects scientifically and technically that still need further improvement for better clarity of this manuscript, I hope authors can make corresponding revisions based on my comments below:

1 Title: "Revisiting the trend in the occurrences of the "warm Arctic-cold Eurasian continent" temperature pattern" Why "revisit"? Have you (authors) done this before? or are there other papers dealing with this matter before? if so, what are the scientific outcome from those existing studies?

2 To my understanding, SOM is a pure advanced statistical tool and there is nothing related to the physics, right? If this is the case, shall I say any results come from SOM have uncertainties because you need to pre-define SOM nodes and this procedure is a kind arbitrary, right? On top of it, as you pointed out in the abstract only 40% of the surface temperature trends are explained by SOM pre-defined nodes that fit to your pre-condition, i.e. warm Arctic-cold Eurasian continent. What I am trying to say is that for what kind of criteria you need to be satisfied before you can make a rebuts conclusion to say: "ok, there is a linkage" or "no, there isn't a linkage". This comment and "a kind of arbitrary" above come from your description on line 141-143.

3 How sensitivity of the data source will impact the final result? In this study, you have applied ERA-Interim data. if you use other data resource, e.g. NCEP or MERRA, would be your conclusion changed entirely or partly? I am not asking to use these data sets to rerun SOM, but it would be nice to comment it at the end of this study.

4 Authors focused on the impacts of the SST anomalies over North Pacific and Atlantic Oceans on the trend in the occurrences of the "warm Arctic cold Eurasian continent" temperature pattern. The influence of decreasing Arctic sea ice cannot be ignored. You may consider to add discussions on the influence of sea ice to your pre-defined warm Arctic and cold Eurasian content.

There are a number of technical details need to be clarified:

a) Fig.1: All "percent" sum together is larger than 100%, please check. b) Fig.2: The color bar refers to what? Contour color? what are the background (fingerprint like) information in each sub-plot? The text explanation for figure 2 (line 182 -185) and figure 2 presentation seems not match to each other. I suggest you remove unnecessary from the plot and only show what you have explained in the text so readers can

**ACPD**
understand better. c) The comment above applied to at least Fig, 3, 4, 5 and 6. d) "same as Figure2, but for,," This is not a good figure caption, please write clear with full information. For those surface fluxes, I think you need to explain the unit of the fluxes, are those daily accumulated fluxes? e) The sea ice concentration figure needs more explanations, e.g. node information was missing; what was meant for positive and negative anomalous? is this also for winter season? how about summer season? Now I realized you actually only investigate winter season for everything, if so, you need to say this explicitly in the beginning of the paper, f) Fig.7 and 14: I have difficult to understand these figures? What we can learn from those figures? If you only tell the integrated total number of days for each node and compared with showing this figure, what we will missing up? g) Fig. 12: "wave activity flux": This need to be explained more in detail both here and in the text. 100m2/s, what is this? and in the caption: 107 m2/s, h) Please mark the study area in corresponding figures 2-6, to help readers understand the mechanism impact more intuitively. i) Table 3 is not mentioned in the article, and some problems of uppercase and lowercase letters (such as not show or Not show), please check them carefully. j) The order of the nodes should be consistent in figures, 10-12. k) Authors should increase some discussions about the application of statistical results in prediction of surface temperature Arctic cold Eurasian continent.

The results in this study are based on statistical analysis. Some numerical experiments may be considered in the further studies.

I hope authors will find my comments useful to make further improvement of this manuscript. The manuscript may need to be reviewed again before it can be accepted for publication in ACP.

---

## Referee Comment (RC2) · Anonymous Referee #2 · 31 Aug 2020

Revisiting the trend in the occurrences of the "warm Arctic-cold Eurasian continent" temperature pattern – review

Recommendation: Minor revisions

**Summary**

This paper investigates the relationship between trends/changes in the occurrence of the warm Arctic-cold Eurasia temperature patterns and numerous atmospheric variables. This paper refutes that this temperature pattern is due to sea-ice melt, and instead suggests that both temperature trends and sea-ice melt are due to cyclical changes in sea-surface temperatures and atmospheric patterns. The authors show

that the warm Arctic-cold Eurasia temperature pattern has been occurring periodically since the mid 1800s and is associated with fluctuations in the AMO, PDO, and Pacific sea-surface temperatures. Overall this paper is well written and figures are clearly described.

**General comments**

The description of the SOM and the transition between nodes is good.

Please refer to figures more throughout the results section. I'd cite the figure number each time you change which figure you are discussing. For example, on line 230 you mention Figure 6, but then in the following line you are referring to Figure 5 but you do not give the figure number. It would be easy here (and in other places) for the reader to be looking at the wrong figure. The paragraph starting at line 277 is another instance where figures should be referred to more frequently.

Datasets and methods section – this section provides a good explanation of SOMs, including what SOMs are and how you will apply them to temperature data, but there is no explanation of how you analyse the other variables (i.e. create composites based on the SOM for temperature data), or the use of principal component analysis. Please include this here.

Consider adding analysis to show what portion of the trend in the warm Arctic-cold Eurasia pattern is due to mean warming. What trend is removed from the 20CR data? It seems an oversight to not consider mean warming when so many other variables are being examined.

**Specific comments**

Lines 23-36 – Abstract nicely sums up the major findings of the paper.

Line 53 – This line states that the warm Arctic-cold continents pattern has been observed on an interannual timescale. Please state here whether the pattern has been strengthening linearly over time, or whether it's a cyclical pattern, or something else.

Line 75 – What changes in the Gulf Stream are you referring to here?

Line 85 – "Using regression method" should probably read "using regression", or "using linear regression" (if this is correct).

Lines 90-98 – This first part of the Datasets and methods section seems to be replicating some of what is said in section 2.2. I'd suggest starting the datasets and methods section with section 2.1, and incorporating lines 90-98 into section 2.2.

Line 94 – Should this say "41 winters"? Or are you only considering complete winters, i.e. December 1979-February 2019 (thus excluding January and February 1979, and December 2019)? Which months do you use for winter? I assume it's DJF.

Line 102 – What is the resolution of the ERA-Interim data?

Lines 137-138 – What dataset are these lines referring to? Both ERA-Interim and 20CR? If both, which 40-year period do you use? I.e. do you subtract the 1979-2019 mean from both datasets?

Line 150 – Do the SOM-explained trends mean something physically, i.e. are they the fraction of the total trends that are explained by changes in circulation (or something else)?

Lines 161-162 – This sentence compares the "first node" in each group, however node 9 appears to be the second node in group one, and node 1 is the first node in group two.

Lines 164-165 – It is not clear from Figure 1 that the maximum anomalies are centered near Svalbard. Please consider adding contour lines to the SOMs, or use a discrete color scale. When you say maximum, are you referring to the greatest departure from zero (i.e. positive or negative values)?

Line 165 – This line states that nodes 3 and 7 are the second most frequently occurring of their groups, but node 3 occurs most frequently. The comparison of pairs is good,

but needs to be worded more carefully. Maybe pick the most frequently occurring node in group 1 then identify its pair.

Lines 171-172 – Why can't this SOM consider temperature trends? I think this should say "does not" not "cannot".

Lines 176-180 – Consider moving these lines to the methods section.

Line 193 – Please add figure reference.

Line 223 – Nice explanation of turbulent heat flux!

Line 229 – Maybe refer back to Figures 2 and 3 if that is where this statement comes from.

Lines 229-230 – Are you sure this is the correct order? I.e. over the Barents Sea in node 1, is it possible that the sea ice melt causes a reduction in the albedo which results in an increased turbulent heat flux?

Line 231 – When you say "larger" do you mean larger spatially, or a greater magnitude anomaly?

Line 238 – "composted" should probably be "composited".

Line 239 – What happens if you do the same lag analysis for sea ice concentration? I think it is important to know that sea ice does not also peak before the day the nodes occur. Similarly, what happens if you do this lag analysis on the geopotential height patterns? It seems strange to say that circulation leads sea ice cover without mentioning the geopotential height patterns.

Lines 250-251 – How does this differ to the other nodes? I assume they only exhibit interannual variability.

Line 255 – I think this should refer to Table 3 (not Table 2).

Line 261 – Figure 8 does not appear to cover a large enough region to determine

whether there are positive trends over southern Europe. This might need re-wording.

Line 262 – Maybe point out that negative trends are mostly not significant.

Line 267 – Arctic–cold should be Arctic-cold

Line 281 – Refer to figure number (Figure 11).

Lines 282-285 – Which node are you referring to? I assume node 1 but this should be clear.

Lines 284-285 – Are you determining the direction of propagation from Figure 11 or Figure 12? From the text it sounds like you are only referring to Figure 11, but I am not sure how you are determining that the Rossby wave moves southeastwards to the Eurasian continent from this figure. Please explain and give figure number.

Lines 285-286 – What figure(s) support the claim that "large SST anomalies over the Nordic Ocean augment the wave signal through local air-sea interaction"? This statement needs more support and/or more of a description on how you came to this conclusion.

Line 290 – Figure number?

Line 302 – Does "these results" refer to the results in Figures 10-12, or to the results you just mentioned in lines 299-302? If you're referring to Figures 10-12, please state this.

Line 308 – Which figure are you referring to here? If this comparison is not shown, write "(not shown)".

Line 321 – Where it states that the magnitude is smaller for the 20 CR data, could this be because the 20 CR data are detrended and the ERA-Interim data are not?

Lines 321-322 – This sentence says "frequencies of all the nodes (Figure 14)", but Figure 14 only shows data for nodes 1, 4, 6, and 9 – please rectify.

Line 322 – Please refer to the corresponding figure that shows node occurrence for ERA-Interim.

Line 325 – The occurrence frequencies at the end of the time series in node 1, Figure 7, appear to be slightly greater than those for node 1 in Figure 14. Could this indicate that mean warming amplifies these trends?

Lines 335-336 – If these results are not shown, please state this.

Lines 343-344 – Why isn't the central North Pacific Ocean SST index shown in Figure 15 since it is significantly correlated with EOF modes 1 and 2?

Line 347 – And the PDO?

Lines 386-387 – Which figures are you referring to here?

Lines 388-389 – How does this atmospheric process suggest that the relationship between a warmer Arctic and East Asian cold spells are not as strong? If the atmospheric patterns described by your SOMs show changes in circulation patterns lead to increases in Arctic temperatures and decreases in Eurasian temperatures, then there appears to be a strong link. Or are you saying that temperature increases in the Arctic are not the driver of temperature decreases in Eurasia?

Figures

In general - Please add the following to the figure captions: - What years the figure covers (if not shown). E.g. Figure 1 - Whether the data have been detrended or not - Dataset used - Consider making figures more consistent, for example, Figure 10 has the Pacific Ocean in the center, whereas Figure 12 has the Atlantic in the center. It would be easier to compare these figures if they both had the same east/west bounds.

Figure 1 - Please consider adding contour lines to the SOM, or use a discrete color scale so it is clearer where the maximum/minimum values are on these plots. - Please mention years and dataset in the caption.

Figure 2 - Please reconsider the use of a rainbow color scale. Reds and greens can look identical to color blind people. - It appears that the stippling/hatching is plotted on top of the contour lines. The plot might be easier to read if the contour lines were on top of the stippling/hatching. - The caption states that this is the "corresponding 500-hPa geopotential height anomalies", but you do not mention that it corresponds to Figure 1. - The caption states that stippled areas are significant, but what about the hatched areas? I assume they are also significant. - Please mention what contour lines show in caption. - Maybe consider rotating the nodes so they match Figure 1 better, i.e. put Russia at the bottom of the subplots. Alternatively, adding an outline of the region in Figure 1 to the plots like Figure 2 would be helpful.

Figure 3 - It would be useful to show the contour lines (from Figure 2) on this plot as well (without stippling) so we can see exactly how the contour lines and wind anomalies line up. - What does the gray shading mean?

Figure 6 - Node numbers are missing from Figure 6. Please add them.

Figure 7 - Consider adding trendlines and p-values to each subplot (and other similar figures).

Figures 10, 11, and 12 - Consider arranging these plots the same, i.e. all 2x2 or 1x4 for easier comparison between the figures.

Figure 14 - Can the results from Figure 7 be overlaid on Figure 14? Maybe with gray dashed outlines. This would make it clearer to see the similarities/differences between the results.

Figure 15 - Consider putting r and p values on subplots b and d. Or in caption.

---

## Author Comment (AC1) · 10 Sep 2020

The linkage between the warn Arctic and mid-latitude weather and climate is a hot topic for cryosphere research community and for this reason, I see this study is interesting and worth to be noticed as a scientific publication. The manuscript is well structured, and the objectives of this study are clear. The content fits well the scope of ACP.

I recommend this manuscript to be published in ACP. However, I see there are some aspects scientifically and technically that still need further improvement for better clarity of this manuscript, I hope authors can make corresponding revisions based on my comments below:

1 Title: "Revisiting the trend in the occurrences of the "warm Arctic-cold Eurasian continent" temperature pattern" Why "revisit"? Have you (authors) done this before? Or are there other papers dealing with this matter before? if so, what are the scientific outcome from those existing studies?

We have not carried out previous research on the potential mechanisms for the trends of warm-Arctic-cold Eurasian per se, but there have been several other studies that are either directly or indirectly related to this specific topic. Two main conclusions regarding the forcing behind the trends stem from these studies. One conclusion is that the recent warm Arctic-cold continents pattern can be attributable to the Arctic sea ice loss (Inoue et al., 2012; Tang et al., 2013; Mori et al., 2014; Kug et al., 2015; Cohen et al., 2018; Mori et al., 2019); The others disputed sea ice loss as a driver for the trend (Blackport et al., 2019; Fyfe, 2019), Instead, they point to internal atmospheric variability and the Pacific and Atlantic SST oscillations as potential forcing behind the trends (Lee et al., 2011; Sato et al., 2014; Matsumura and Kosaka, 2019; Clark and Lee, 2019). Most of these previous studies and the two school of thought were mentioned in the Introduction. Our work, which took a different approach, confirmed the second school of thought. Because of these existing studies on this topic, we used the word 'revisiting' in the title of our manuscript.

2 To my understanding, SOM is a pure advanced statistical tool and there is nothing related to the physics, right? If this is the case, shall I say any results come from SOM have uncertainties because you need to pre-define SOM nodes and this procedure is a kind arbitrary, right? On top of it, as you pointed out in the abstract only 40% of the surface temperature trends are explained by SOM pre-defined nodes that fit to your pre-condition, i.e. warm Arctic-cold Eurasian continent. What I am trying to say is that for what kind of criteria you need to be satisfied before you can make a rebuts conclusion to say: "ok, there is a linkage" or "no, there isn't a linkage". This comment and "a kind of arbitrary" above come from your description on line 141-143.

SOM is an advanced statistical tool for pattern extraction. Although SOM is superior to some other existing pattern extraction tools such as EOF, it suffers from the same limitations as other statistical tools in identifying physical modes. That was why a large part of the manuscript was devoted to explain the existence of the patterns and their trends based on physical understanding of atmosphere and ocean dynamics that had been established from theoretical framework and/or from coupled ocean-atmosphere modeling. Yes, to use the SOM method, one has to pre-define SOM nodes and the procedure is not completely objective. A small grid (each node has larger frequency of occurrence) tends to miss transitions between the main patterns that are retained by a large grid. But an excessively large grid could

sidetrack the attention from the main variability patterns. Nevertheless, changing the grid from 3x3 to 4x4 or even larger would not change the main conclusion.

3 How sensitivity of the data source will impact the final result? In this study, you have applied ERA-Interim data. if you use other data resource, e.g. NCEP or MERRA, would be your conclusion changed entirely or partly? I am not asking to use these data sets to rerun SOM, but it would be nice to comment it at the end of this study.

We believe our results are not particularly sensitive to the specific large-scale reanalysis data source. We could have also used ERA5, or NCEP or MERRA and arrived at similar conclusions, although there might be some minor differences. We have added some comments on this point at the end of the study.

4 Authors focused on the impacts of the SST anomalies over North Pacific and Atlantic Oceans on the trend in the occurrences of the "warm Arctic cold Eurasian continent" temperature pattern. The influence of decreasing Arctic sea ice cannot be ignored.
You may consider to add discussions on the influence of sea ice to your pre-defined warm Arctic and cold Eurasian content.

We added some discussions on the influence of sea ice in the Conclusions and Discussions section.

There are a number of technical details need to be clarified:
a) Fig.1: All "percent" sum together is larger than 100%, please check.
Changed

b) Fig.2: The color bar refers to what? Contour color? what are the background (fingerprint like) information in each sub-plot? The text explanation for figure 2 (line 182 -185) and figure 2 presentation seems not match to each other. I suggest you remove unnecessary from the plot and only show what you have explained in the text so readers can understand better.
Both color bar and contour color refer to 500-hPa geopotential height anomalies. Dotted regions in each sub-plot indicate the above 95% confidence level.
We revised some of the discussion.

c) The comment above applied to at least Fig, 3, 4, 5 and 6.
In Figure 3-6, shaded and dotted regions indicate the above 95% confidence level.

d) "same as Figure2, but for,,," This is not a good figure caption, please write clear with full information. For those surface fluxes, I think you need to explain the unit of the fluxes, are those daily accumulated fluxes?
We revised figure caption with details. The fluxes are daily accumulated fluxes, which are now explained in the caption and text.

e) The sea ice concentration figure needs more explanations, e.g. node information was missing; what was meant for positive and negative anomalous? is this also for winter season? how about summer season? Now I realized you actually only investigate winter season for

everything, if so, you need to say this explicitly in the beginning of the paper.

We added node information. The anomalous sea ice concentration is a composite result based on the occurrences of nodes. For example, the negative sea ice concentration corresponds to the spatial pattern of air temperature for node 1. In this paper, we only examine warm Arctic-cold continents pattern in boreal winter, which was mentioned in the first and second paragraph of the manuscript.

f) Fig.7 and 14: I have difficult to understand these figures? What we can learn from those figures? If you only tell the integrated total number of days for each node and compared with showing this figure, what we will missing up?

Figure 7 and 14 show the integrated total number of days for each node. In Figure 7 and 14, the numbers for nodes 1 and 4 are larger after 2000 than those prior to 2000. The opposite occurs for nodes 6 and 9. Figure 14 mainly show an interdecadal variability of the number. The trends in the number for nodes 1, 4, 6, and 9 are a fragment of the interdecadal variability. We added clarification in the discussion.

g) Fig. 12: "wave activity flux": This need to be explained more in detail both here and in the text. 100m2/s, what is this? and in the caption:107 m2/s.

"vector $100m^2/s$" in the figure is figure legend of wave activity flux. The unit of stream function is $m^2/s$ and its magnitude is the product of the values in the figure and $10^7$. We have added explanation of wave activity flux in the discussion and in the figure caption with a reference.

h) Please mark the study area in corresponding figures 2-6, to help readers understand the mechanism impact more intuitively.

Marked

i) Table 3 is not mentioned in the article, and some problems of uppercase and lowercase letters (such as not show or Not show), please check them carefully.

Changed

j) The order of the nodes should be consistent in figures, 10-12.

Changed

k) Authors should increase some discussions about the application of statistical results in prediction of surface temperature Arctic cold Eurasian continent.

Added discussion

The results in this study are based on statistical analysis. Some numerical experiments may be considered in the further studies.

Added

---

## Author Comment (AC2) · 10 Sep 2020

General comments

The description of the SOM and the transition between nodes is good.

Please refer to figures more throughout the results section. I'd cite the figure number each time you change which figure you are discussing. For example, on line 230 you mention Figure 6, but then in the following line you are referring to Figure 5 but you do not give the figure number. It would be easy here (and in other places) for the reader to be looking at the wrong figure. The paragraph starting at line 277 is another instance where figures should be referred to more frequently.

Good suggestion. We have gone through the manuscript carefully and added citations to figures whenever appropriate.

Datasets and methods section – this section provides a good explanation of SOMs, including what SOMs are and how you will apply them to temperature data, but there is no explanation of how you analyse the other variables (i.e. create composites based on the SOM for temperature data), or the use of principal component analysis. Please include this here.

Thanks for pointing out this oversight. We have added more description about the other methods we also used in the analyses, in addition to SOM, in the Method section.

Consider adding analysis to show what portion of the trend in the warm Arctic-cold Eurasia pattern is due to mean warming. What trend is removed from the 20CR data? It seems an oversight to not consider mean warming when so many other variables are being examined.

Trend in wintertime surface air temperature anomalies for the 1854-2014 period for the 20CR data was removed.

In this study, we mainly focused on the role of the interdecadal variability of SST anomalies over northern oceans in trend in the warm Arctic-cold Eurasia pattern. In Conclusions and Discussions section, we increased some discussions of the role of Arctic warming in the trend.

Specific comments

Lines 23-36 – Abstract nicely sums up the major findings of the paper.

Thanks

Line 53 – This line states that the warm Arctic-cold continents pattern has been observed on an interannual timescale. Please state here whether the pattern has been strengthening linearly over time, or whether it's a cyclical pattern, or something else.

We have added a statement here about increasing trend in the occurrence of the warm Arctic-cold continents pattern.

Line 75 – What changes in the Gulf Stream are you referring to here?

Changed the statement to "… the sea surface temperature anomalies over the Gulf Stream."

Line 85 – "Using regression method" should probably read "using regression", or "using linear regression" (if this is correct).

Changed to 'using linear regression'

Lines 90-98 – This first part of the Datasets and methods section seems to be replicating some of what is said in section 2.2. I'd suggest starting the datasets and methods section with section 2.1, and incorporating lines 90-98 into section 2.2.

Removed the replications

Line 94 – Should this say "41 winters"? Or are you only considering complete winters, i.e. December 1979-February 2019 (thus excluding January and February 1979, and December 2019)? Which months do you use for winter? I assume it's DJF.

Winter is defined by DJF and we only consider complete winters from December 1979 through February 2019. This is now clarified.

Line 102 – What is the resolution of the ERA-Interim data?

The resolution of the ERA-Interim was added.

Lines 137-138 – What dataset are these lines referring to? Both ERA-Interim and 20CR? If both, which 40-year period do you use? I.e. do you subtract the 1979-2019 mean from both datasets?

These lines refer to ERA-Interim reanalysis. We subtract the 1979-2019 mean from ERA-Interim.

Line 150 – Do the SOM-explained trends mean something physically, i.e. are they the fraction of the total trends that are explained by changes in circulation (or something else)?

The SOM-explained trends are the fraction of the total trends that are explained by the changes in circulations.

Lines 161-162 – This sentence compares the "first node" in each group, however node 9 appears to be the second node in group one, and node 1 is the first node in group two.

Changed

Lines 164-165 – It is not clear from Figure 1 that the maximum anomalies are centered near Svalbard. Please consider adding contour lines to the SOMs, or use a discrete color scale. When you say maximum, are you referring to the greatest departure from zero (i.e. positive or negative values)?

Contour lines are added. Maximum refers to largest values of the anomalies

Line 165 – This line states that nodes 3 and 7 are the second most frequently occurring of their groups, but node 3 occurs most frequently. The comparison of pairs is good, but needs to be worded more carefully. Maybe pick the most frequently occurring node in group 1 then identify its pair.

Good suggestion. Statements rephrased.

Lines 171-172 – Why can't this SOM consider temperature trends? I think this should say "does not" not "cannot".

Changed to "does not"

Lines 176-180 – Consider moving these lines to the methods section.

We have added some description on composite method in the Method section, following another reviewer's comment.

Line 193 – Please add figure reference.

Referred more to figures whenever appropriate.

Line 223 – Nice explanation of turbulent heat flux!

Thanks

Line 229 – Maybe refer back to Figures 2 and 3 if that is where this statement comes from.

Made references back to the figures

Lines 229-230 – Are you sure this is the correct order? I.e. over the Barents Sea in node 1, is it possible that the sea ice melt causes a reduction in the albedo which results in an increased

turbulent heat flux?

We believe the cause-effect is correct based on previous studies (Blackport et al., 2019)

Line 231 – When you say "larger" do you mean larger spatially, or a greater magnitude anomaly?

A greatermagnitude anomaly.    Clarified

Line 238 – "composted" should probably be "composited".

Changed

Line 239 – What happens if you do the same lag analysis for sea ice concentration? I think it is important to know that sea ice does not also peak before the day the nodes occur. Similarly, what happens if you do this lag analysis on the geopotential height patterns?

It seems strange to say that circulation leads sea ice cover without mentioning the geopotential height patterns.

The pattern of the composited anomalous 500-hPa geopotential height, turbulent heat flux, and sea ice concentration 2 days prior to the day when the nodes occur (not shown) is similar to the simultaneous pattern in Figures 2, 5, and 6.

Lines 250-251 – How does this differ to the other nodes? I assume they only exhibit interannual variability.

The main difference is the decadal variability.

Line 255 – I think this should refer to Table 3 (not Table 2).

Changed

Line 261 – Figure 8 does not appear to cover a large enough region to determine whether there are positive trends over southern Europe. This might need re-wording.

Rewording done

Line 262 – Maybe point out that negative trends are mostly not significant.

Done

Line 267 – Arctic–cold should be Arctic-cold

Changed

Line 281 – Refer to figure number (Figure 11).

Added reference to Figure 11

Lines 282-285 – Which node are you referring to? I assume node 1 but this should be clear.

Added reference to node 1.

Lines 284-285 – Are you determining the direction of propagation from Figure 11 or Figure 12? From the text it sounds like you are only referring to Figure 11, but I am not sure how you are determining that the Rossby wave moves southeastwards to the Eurasian continent from this figure. Please explain and give figure number.

The direction of wave activity flux points to the Eurasian continent (Figure 12). A reference to Figure 12 is added.

Lines 285-286 – What figure(s) support the claim that "large SST anomalies over the Nordic Ocean augment the wave signal through local air-sea interaction"? This statement needs more support and/or more of a description on how you came to this conclusion.

Added more descriptions with reference to figures

Line 290 – Figure number?

Added

Line 302 – Does "these results" refer to the results in Figures 10-12, or to the results you just

mentioned in lines 299-302? If you're referring to Figures 10-12, please state this.

Reference to Figures 10-12 are added

Line 308 – Which figure are you referring to here? If this comparison is not shown, write "(not shown)".

"(not shown)" was added.

Line 321 – Where it states that the magnitude is smaller for the 20 CR data, could this be because the 20 CR data are detrended and the ERA-Interim data are not?

Added detrending of the 20CR as a potential explanation

Lines 321-322 – This sentence says "frequencies of all the nodes (Figure 14)", but Figure 14 only shows data for nodes 1, 4, 6, and 9 – please rectify.

Clarified

Line 322 – Please refer to the corresponding figure that shows node occurrence for ERA-Interim.

Reference to corresponding figures added

Line 325 – The occurrence frequencies at the end of the time series in node 1, Figure 7, appear to be slightly greater than those for node 1 in Figure 14. Could this indicate that mean warming amplifies these trends?

Global warming may be a reason

Lines 335-336 – If these results are not shown, please state this.

Stated

Lines 343-344 – Why isn't the central North Pacific Ocean SST index shown in Figure 15 since it is significantly correlated with EOF modes 1 and 2?

The central North Pacific Ocean SST index is added in Figure 15

Line 347 – And the PDO?

Added

Lines 386-387 – Which figures are you referring to here?

References to corresponding figure added

Lines 388-389 – How does this atmospheric process suggest that the relationship between a warmer Arctic and East Asian cold spells are not as strong? If the atmospheric patterns described by your SOMs show changes in circulation patterns lead to increases in Arctic temperatures and decreases in Eurasian temperatures, then there appears to be a strong link. Or are you saying that temperature increases in the Arctic are not the driver of temperature decreases in Eurasia?

Temperature increases in the Arctic are not the driver of temperature decreases in Eurasia.

Figures

In general - Please add the following to the figure captions: - What years the figure covers (if not shown). E.g. Figure 1 - Whether the data have been detrended or not -

Dataset used - Consider making figures more consistent, for example, Figure 10 has the Pacific Ocean in the center, whereas Figure 12 has the Atlantic in the center. It would be easier to compare these figures if they both had the same east/west bounds.

Years and data were added in figure captions. Figure 10 has changed.

Figure 1 - Please consider adding contour lines to the SOM, or use a discrete color scale so it is clearer where the maximum/minimum values are on these plots. – Please mention years and

dataset in the caption.

Figure 1 has been changed into contour lines.

Figure 2 - Please reconsider the use of a rainbow color scale. Reds and greens can look identical to color blind people. - It appears that the stippling/hatching is plotted on top of the contour lines. The plot might be easier to read if the contour lines were on top of the stippling/hatching. - The caption states that this is the "corresponding 500-hPa geopotential height anomalies", but you do not mention that it corresponds to Figure 1. - The caption states that stippled areas are significant, but what about the hatched areas? I assume they are also significant. - Please mention what contour lines show in caption. - Maybe consider rotating the nodes so they match Figure 1 better, i.e. put Russia at the bottom of the subplots. Alternatively, adding an outline of the region in Figure 1 to the plots like Figure 2 would be helpful.

Rainbow color scale is now used. An outline of the region in Figure 1 is added. We used stippled, not hatched in Figure 2.

Figure 3 - It would be useful to show the contour lines (from Figure 2) on this plot as well (without stippling) so we can see exactly how the contour lines and wind anomalies line up. - What does the gray shading mean?

Adding contour lines made it harder to see vectors. We replaced stipping by shading to denote the above 95% confidence level.

Figure 6 - Node numbers are missing from Figure 6. Please add them.

Added

Figure 7 - Consider adding trend lines and p-values to each subplot (and other similar figures).

Added

Figures 10, 11, and 12 - Consider arranging these plots the same, i.e. all 2x2 or 1x4 for easier comparison between the figures.

Rearranged

Figure 14 - Can the results from Figure 7 be overlaid on Figure 14? Maybe with gray dashed outlines. This would make it clearer to see the similarities/differences between the results.

The time series in Figure 7 is added in Figure 14

Figure 15 - Consider putting r and p values on subplots b and d. Or in caption.

R and P values are added in the caption

---

## Author Comment (AC3) · 10 Sep 2020

we uploaded the revised manuscript as figures

[Figure]

**Revisiting the trend in the occurrences of the "warm Arctic-cold Eurasian continent"**

**temperature pattern**

Lejiang Yu[1,2]*, Shiyuan Zhong[3], Cuijuan Sui[4] , and Bo Sun[1]

1MNR Key Laboratory for Polar Science, Polar Research Institute of China, Shanghai, China

2 Southern Marine Science and Engineering Guangdong Laboratory (Zhuhai), Zhuhai, Guangdong,

China

3Department of Geography, Environment and Spatial Sciences, Michigan State University, East

Lansing, MI, USA

4 National Marine Environmental Forecasting Center, Beijing, China

*Corresponding Author's address

**Revisiting the trend in the occurrences of the "warm Arctic-cold Eurasian continent"**

                        **temperature pattern**

Lejiang Yu[1,2]*, Shiyuan Zhong[3], Cuijuan Sui[4] , and Bo Sun[1]

1MNR Key Laboratory for Polar Science, Polar Research Institute of China, Shanghai, China

2 Southern Marine Science and Engineering Guangdong Laboratory (Zhuhai), Zhuhai, Guangdong,

China

3Department of Geography, Environment and Spatial Sciences, Michigan State University, East

Lansing, MI, USA

4 National Marine Environmental Forecasting Center, Beijing, China

*Corresponding Author's address

---

## Author Response (AR2)

The figure captions can still be improved, e.g.

Fig.2: the sub-domain in 1 is the study area. Then you don't need to repeat in other figures caption.

We deleted the sentence in the figure captions except for Figure 2.

In many figures caption, you stated the same text: "without removing,,", you can write it in the text

We deleted the expression, and added a sentence at the end of the first paragraph in the section Datasets and methods ("
[revised manuscript text omitted]